# BAERLIN2014 – stationary measurements and source apportionment at an urban background station in Berlin, Germany

Erika von Schneidemesser[1], Boris Bonn[1*], Tim M. Butler[1], Christian Ehlers[2α], Holger Gerwig[3], Hannele Hakola[4], Heidi Hellén[4], Andreas Kerschbaumer[5], Dieter Klemp[2], Claudia Kofahl[2β], Jürgen Kura[3], Anja Lüdecke[3], Rainer Nothard[5], Axel Pietsch[3], Jörn Quedenau[1], Klaus Schäfer[6], James J. Schauer[7], Ashish Singh[1], Ana-Maria Villalobos[7], Matthias Wiegner[8], Mark G. Lawrence[1]

[1]Institute for Advanced Sustainability Studies (IASS), D-14467 Potsdam, Germany
[2]IEK-8, Research Centre Jülich, D-52425 Jülich, Germany
[3]Division Environmental Health and Protection of Ecosystems, German Environment Agency, D-06844 Dessau-Roßlau, Germany
[4]Finnish Meteorological Institute, FI-00560 Helsinki, Finland
[5]Senate Department for the Environment, Transport and Climate Protection, D-10179 Berlin, Germany
[6]Institute of Meteorology and Climate Research, Atmospheric Environmental Research (IMK-IFU), Karlsruhe Institute of Technology (KIT), D-82467 Garmisch-Partenkirchen, Germany
[7]Environmental Chemistry and Technology Program, University of Wisconsin-Madison, Madison 53705, WI, USA
[8]Ludwig-Maximilians-Universität, Meteorological Institute, D-80333 Munich, Germany
*now at: Chair of Ecosystem Physiology, Institute of Forest Sciences, Albert-Ludwig Universität, D-79110 Freiburg, Germany
αnow at: Fachbereich 42: Kontinuierliches Luftqualitätsmessnetz, Landesamt für Natur, Umwelt und Verbraucherschutz NRW, D-45133 Essen, Germany
βnow at: Institut für Physikalische Chemie, Georg-August-Universität, D-37077 Göttingen, Germany

*Correspondence to Erika von Schneidemesser (evs@iass-potsdam.de)*

**Abstract.** The Berlin Air quality and Ecosystem Research: Local and long-range Impact of anthropogenic and Natural hydrocarbons (BAERLIN2014) campaign was conducted during the three summer months (June-August) of 2014. During this measurement campaign, both stationary and mobile measurements were undertaken to address complementary aims. This paper provides an overview of the stationary measurements and results that were focused on characterization of gaseous and particulate pollution, including source attribution, in the Berlin-Potsdam area, and quantification of the role of natural sources in determining levels of ozone and related gaseous pollutants. Results show that biogenic contributions to ozone and particulate matter are substantial. One indicator for ozone formation, the OH reactivity, showed a 31% ($0.82 \pm 0.44$ s$^{-1}$) and 75% ($3.7 \pm 0.90$ s$^{-1}$) contribution from biogenic NMVOCs for urban background ($2.6 \pm 0.68$ s$^{-1}$) and urban park ($4.9 \pm 1.0$ s$^{-1}$) location, respectively, emphasizing the importance of such locations as sources of biogenic NMVOCs in urban areas. A comparison to NMVOC measurements made in Berlin approx. 20 years earlier generally show lower levels today for anthropogenic NMVOCs. A substantial contribution of secondary organic and inorganic aerosol to $PM_{10}$ concentrations was quantified. In addition to secondary aerosols, source apportionment analysis of the organic carbon fraction identified the contribution of biogenic (plant-based) particulate matter, as well as primary contributions from vehicles, with a larger contribution from diesel compared to gasoline vehicles, as well as a relatively small contribution from wood burning, linked to measured levoglucosan.

## 1    Introduction

Air pollution and climate change are two of the most prescient environmental problems of our age. Recent research from the Global Burden of Disease study and others attribute over 3 million premature deaths to outdoor air pollution globally in 2013 (Brauer et al., 2016;Lelieveld et al., 2015;WHO, 2016). A report by the World Bank (WorldBank, 2016) estimated the 2013 welfare losses owing to ambient surface level $PM_{2.5}$ and $O_3$ air pollution to be equivalent to 5% of GDP in Europe, and often more in other world regions. Studies have shown that a changing climate will exacerbate ozone owing to increased temperatures and other factors, such as additional meteorological parameters and less effective emissions controls, that are favorable to ozone formation (Jacob and Winner, 2009;Rasmussen et al., 2013). One such factor is a projected increase in biogenic volatile organic compound emissions, such as isoprene or monoterpenes. While these increases are expected to be compensated for by much larger declines in anthropogenic emissions, as also indicated in other studies e.g. Colette et al. (2013) or West et al. (2013), there are additional impacts that are not yet captured by the models, such as those of secondary organic aerosol (SOA) among others, that show that such estimates of climate change effects are likely underestimated (Geels et al., 2015). While significant reductions in $O_3$ precursor emissions have been observed over the past couple decades, and peak ozone levels have been declining over much of north-western Europe, a comparable reduction in mean ozone has not followed (Derwent, 2008;Ehlers et al., 2016). This is particularly relevant for countries where the majority of the population resides in cities. In Europe during 2012-2014, more than 85% of the urban population has been exposed to air pollutant concentrations of ozone and $PM_{2.5}$ exceeding the recommended WHO limit values for the protection of human health, as well as substantial exceedances at the roadside of nitrogen dioxide ($NO_2$) (EEA, 2016). In this context, it is crucial that we further improve our understanding of the sources of air pollutants in urban areas, as well as the contribution of natural sources to secondary pollutants such as ozone. Furthermore, recent research has shown that chemical products are emerging as the largest sources of non-methane volatile organic compounds in urban areas, owing to the previous regulatory focus on transport emissions (McDonald et al., 2018). An improved understanding of sources will allow for approaches that can better target the most relevant sources for mitigation, as well as accounting for the linkages between air quality and climate change in developing strategies for action on climate change and the reduction of air pollution, to improve health and create more livable cities.

The Berlin Air quality and Ecosystem Research: Local and long-range Impact of anthropogenic and Natural hydrocarbons 2014 (BAERLIN2014) campaign aimed to address some of these issues in the context of the Berlin-Potsdam urban area. The campaign had three main aims, (1) characterization of gaseous and particulate pollution, including source attribution, in the Berlin-Potsdam area, (2) quantification of the role of natural sources, specifically vegetation, in determining levels of gaseous pollutants, specifically ozone, and (3) improved understanding of the heterogeneity of pollutants throughout the city. In this paper, only aims (1) and (2) will be addressed. An overview paper describing the mobile measurements, which focused more on aim (3) was published previously (see Bonn et al. (2016)). Because of the focus on ozone and secondary pollutant formation, the campaign was conducted during the three summer months (June-August) of 2014, i.e. the time of maximum ozone pollution levels. Furthermore, while the mobile measurements covered the larger Berlin-Potsdam area, the stationary measurements were focused on an urban background location within the center of Berlin.

The unique characteristics of Berlin were particularly relevant to this study, in that it is a large urban area (population approx. 3.5 million) with significant vegetation. Of the approx. 890 km$^2$ that Berlin covers, approx. 34% of the land surface area is covered by vegetated areas and 6% by water (Senatsverwaltung für Stadtentwicklung III F, 2010). An existing air quality monitoring network (in German: Berliner Luftgüte Messnetz, abbreviated BLUME) provided data on which the campaign could build and leverage. Data from the 16 stations that comprised the BLUME network showed that the EU 8-hour ozone target value of 120 µg m$^{-3}$ was exceeded 12-13 times at each of the two urban background stations that measure ozone (MC010 & MC042) and between 12-21 times per station at the stations on the periphery of the city (referred to here as Berlin rural stations) in 2014 (Stülpnagel et al., 2015). Six of these exceedances in the urban background occurred during the BAERLIN2014 campaign. Furthermore, the regulatory limit value for annual NO$_2$ of 40 µg m$^{-3}$ was exceeded at all six roadside stations in 2014, and although the annual PM$_{10}$ limit value was met, four out of five traffic stations where PM$_{10}$ was measured also exceeded the daily limit value of 50 µg m$^{-3}$ more than the allowed 35 times; the exceedances at the urban background and Berlin rural stations ranged from 14 to 34 times (Stülpnagel et al., 2015). In short, the issue of air pollution has been recognized in Berlin as being in need of action. In this paper, we focus on the stationary measurements conducted at the urban background site in the Berlin city center. A brief overview is given of the suite of measurements conducted and the results obtained. This is followed by more detailed analysis of (1) the NMVOC data and the role in ozone formation including a comparison to a previous study in London and Paris (von Schneidemesser et al., 2011), as well as other urban areas, and (2) source apportionment analysis of PM$_{10}$ filter samples, including a rough comparison of the results to existing emission inventories.

## 2    Methods

A complete list of the parameters measured and their associated instrument descriptions are summarized in Table 1.

### 2.1 Site description

The monitoring station that was the basis for the stationary measurements during the BAERLIN2014 campaign was AirBase station DEBE034, which is maintained as part of the Berlin air quality measurement network (BLUME; BLUME network code MC042), and was located at the corner of Nansenstrasse and Framstrasse in the Neukölln district, in southeast central Berlin (52° 29' 21,98" N, 13° 25' 51,08" E) in a predominantly residential neighborhood, as shown in Figure 1. The station was located on the street corner next to a kindergarten and was classified as an urban background station. According to the location placement dictated by the EU Directive definition (EC, 2008), locations that are situated away from any strong point sources including major roads, typically in a residential neighborhood, but still in the urban core influenced by all sources upwind of the station are classified as urban background. These sites should in theory be representative of the general levels of pollution observed in a city and are used to assess exposure of the general population to air pollutants. This station will likely experience a comparatively high fraction of traffic-related emissions, since some fairly large inner-city thoroughfares were located within a 1 km radius of the site, but as appropriate for an urban background station will not be dominated by traffic like a site located at a major intersection. In addition, a measurement van was used to augment the capacity of the measurement station and

was located approximately 5 meters from the station, parked at the curb of the street (see Figure 1). Finally,
owing to the presence of taller trees in that part of city, including in the vicinity of the monitoring station, one
instrument (ceilometer) was located on the roof of the kindergarten to achieve an unobstructed view skywards,
approximately 5 meters on the opposite side of the measurement station to the van.
A number of NMVOC canister samples were taken in locations throughout the city as part of the mobile
measurements that augmented the stationary measurements in Neukölln. A subset of these were included in the
companion paper to this one covering the mobile measurements (Bonn et al., 2016). These sites where multiple
NMVOC canister samples were taken include Altlandsberg, Plänterwald, the Tiergarten Tunnel, and the so-
called 'AVUS Motorway' during a traffic jam. Further details to the sampling environment can be found in
Table 2. For more information on locations and/or sampling, see also Bonn et al. (2016).

### 2.2 Instrument descriptions

Complementing the BLUME measurements (see (Stülpnagel et al., 2015) or (Geiß et al., 2017) for
details) were additional $PM_{10}$ filter samples collected for elemental carbon (EC) and organic carbon (OC), ions,
and organic tracer analysis; intermittent canister and cartridge samples for the quantification of non-methane
volatile organic compounds (NMVOCs) from an inlet next to the $PM_{10}$ inlet on the roof of the measurement
station; a quadrupole Proton Transfer Reaction Mass Spectrometer (high sensitivity PTR-MS, Ionicon) up in the
van for the measurement of NMVOCs; a set of particle instruments to measure number concentration, size
distribution and surface area also located in the van (section 2.2.4); and a ceilometer CL51 (Vaisala GmbH,
Hamburg) situated on the roof of the kindergarten. A complete list of instruments, parameters measured, and
references for the methods used are provided in Table 1. Further details for the NMVOC measurements are
provided in Table S1. Additional information is provided below.

### 2.2.1    NMVOC Canister Samples

The canisters were prepared to remove ozone using a heated silco-steel capillary (120 °C) prior to
sampling. The cylinders were then pressurized using synthetic air to reduce the relative humidity of the sample.
All NMVOC canister samples taken at Neukölln had a 20 minute sampling duration. After sampling, the
canisters were promptly shipped to FZJ for analysis by GC-FID-MS and were analyzed with no more than five
days between sampling and analysis. Analysis was done using a gas chromatographic system based on a
conventional gas chromatograph (Agilent 6890) equipped with a flame ionization detector (FID), and a mass
spectrometer (Agilent 5975C MSD) for the identification of the trace species. To analyze VOCs at trace gas
levels, a cryogenic pre-concentration was used, consisting of a sample loop (silco steel, 20 cm length, inner
diameter 2 mm) which was cooled down with cold gas above liquid nitrogen (see also Figure 14 in Ehlers et al.,
(2016)). A volume of 800 mL was pre-concentrated in the sample loop at a flow of 80 mL min$^{-1}$.
Subsequently, the sample was thermally desorbed at 120° C and injected on a capillary column (DB-1,
120 m, 0.32 mm ID, 3μm film thickness). After injection, the column was kept isothermal at -60°C for 5 min,
then heated to 200° C at a rate of 4° min$^{-1}$ and finally maintained at 220° C for 10 min. Signals were gathered
from a flame ionization detector and a MSD, which each received 50% of the column output through a split
valve. Analysis of one sample lasted for about 90 min, and sets of 10 cylinders (stainless steel canister, volume:
6 L, Supelco Co., Bellefonte, PA, USA) could be analyzed by unattended operation.
The impact of canister transport and storage was assessed: $C_2$ - $C_{11}$ alkanes, alkenes and aromatic
compounds were found to be stable within 5% over three days compared with an instantaneously analysed
sample. Oxygenated compounds differed by up to 10% and terpenes by up to 20% over the same time period
(Hengst, 2007). In addition, measurement accuracy depends on the uncertainty of the calibration standard (< 5%
between true and declared gas concentrations, Apel-Riemer Environmental Inc.) and that of the mass-flow
controller (< 2% deviation, MKS Instruments, Wilmington, MA, USA). Integration uncertainties ($\Delta\mu$VOC) of
the peak areas were dependent on their respective detection limits ($DL_i$), which are estimated as in equation 1.
$$\Delta\mu VOCi \approx \begin{cases} DL_i & \text{for } \mu VOC_i \text{ nextto } DL_i \\ (0,03-0,06)*\mu VOC_i & \text{otherwise} \end{cases} \qquad (1)$$

Apart from concentrations and their respective detection limits geometrical addition of all these factors yielded
overall experimental uncertainties of less than 10% (for a detailed discussion refer to Urban (2010)).
**2.2.1.1  Canister Samples and OH Reactivity Calculations**
While a total of 103 compounds were quantified by GC-MS in the canister samples, not all of those
compounds were regularly detected in the samples. Furthermore, to be able to make reasonable comparisons
with previous work regarding the contribution of different compound classes to the measured mixing ratios of
NMVOCs, as well as the OH reactivity attributed to these NMVOCs, a subset of the compounds was selected
and used in the analysis. This subset was based on a number of papers in the literature that were also done in
urban areas, and those compounds that were regularly included in OH reactivity calculations (e.g. (Dolgorouky
et al., 2012;Gilman et al., 2009;Goldan et al., 2004;Liu et al., 2008)). This includes 57 NMVOCs (see SI).
Furthermore, even if all compounds were included, there would still be missing reactivity that is not captured
and because no OH measurements were made, the amount of missing reactivity cannot be reliably quantified,
therefore the measured OH reactivity here is a lower limit. Owing to an undetermined source of contamination at
the urban background site, the measurement of n-butane was compromised, and was therefore not included
among the NMVOCs despite typically being reported in the literature. The data subsequently presented in this
paper from the canister samples includes only these 57 compounds unless otherwise noted. For a complete list of
the 103 compounds measured in the samples, including the concentrations reported for a subset of the samples
discussed here, please see Bonn et al. (2016).
A number of canister samples were taken at different locations throughout the city, some with multiple
measurements and some single samples. Five locations had multiple samples, including the main measurement
site at the urban background station (DEBE034) in Neukölln (n=18), Plänterwald (n=11), Altlandsberg (n=10),
the Tiergarten Tunnel (n=9), and the AVUS motorway during a traffic jam (n=2). All samples were taken during
the month of August, will all samples except those in Neukölln taken on one day for any given location (Bonn et
al., 2016).  The samples in the Tiergarten tunnel and on the motorway are most indicative of NMVOC emissions
from traffic.
**2.2.2    NMVOC Cartridge Samples**
NMVOCs (aromatic hydrocabons, terpenes, $C_6$-$C_{10}$ alkanes) were collected into stainless steel
cartridges (6.3 mm ED x 90 mm, 5.5 mm ID) filled with Tenax-TA (60/80 mesh, Supelco, Bellafonte, USA) and
Carboback-B (60/80 mesh, Supelco, Bellafonte, USA) by using a flow rate of 100 ml min$^{-1}$ with a sampling time
of 1 - 4.5 h (Mäki et al., 2017). To prevent the degradation of BVOC by $O_3$, a catalyst heated to 150ºC was used.

Individual VOCs were identified and quantified using a thermal desorption instrument (Perkin-Elmer
TurboMatrixTM 650, Waltham, USA) connected to a gas chromatograph (Perkin-Elmer® Clarus® 600,
Waltham, USA) with a DB-5MS (60 m, 0.25 mm, 1 µm) column and a mass selective detector (Perkin-Elmer®
Clarus® 600T, Waltham, USA). Five-point calibration was utilised using liquid standards in methanol solutions.
Standard solutions were injected onto adsorbent tubes that were flushed with nitrogen (HiQ $N_2$ 6.0 >99.9999%,
Linde AG, Pullach, Germany) flow (100 ml min$^{-1}$) for 10 min in order to remove methanol. For aromatic
hydrocarbons (benzene, toluene, ethylbenzene, p/m-xylene, styrene, o-xylene, propylbenzene, ethyltoluenes,
trimethylbenzenes) detection limits (LODs) varied between 5 and 60 ng m$^{-3}$ , for $C_{6-10}$ alkanes (hexane, heptane,
octane, nonane, decane) between 5 and 10 ng m$^{-3}$ and for isoprene LOD was 21 ng m$^{-3}$. The quantified
monoterpenes (MT) were α-pinene, camphene, β-pinene, $\Delta^3$-carene, p-cymene, limonene, 1,8-cineol, nopinone,
terpinolene and bornylacetate with limit of detection in the range of 3-17 ng m$^{-3}$; sesquiterpenes were
longicyclene, iso-longifolene, aromadendrene, β-caryophyllene and α-humulene with LOD of 20 ng m$^{-3}$.

**2.2.3    NMVOC PTR-MS Measurements**
In addition to canister and cartridge samples, NMVOCs were continuously measured over time by a high-
sensitivity proton transfer reaction mass spectrometer (PTR-MS, Ionicon, built in 2008) (Lindinger et al., 1993).
In brief NMVOCs with a higher proton affinity than water vapor were charged via $H_3O^+$ ions and subsequently
mass selectively detected by applying a distinct electric field strength for the individual masses selected. More
details on the techniques can be found elsewhere (Blake et al., 2009). In total, 72 selected NMVOCs were
measured between June 11 and August 29, 2014 via a heated inlet (T = 60°C) at street level out of the street
facing window of a measurement van (MW088) at approximately 2.5 m above surface. Note that this PTR-MS
detected integer ion mass numbers only and no time of flight option was available for this version. Selection of
masses were based on two aspects: first, typical mass to charge (m/z) ratios for anthropogenic and biogenic
sources like benzene, toluene, isoprene and terpenes, and second, on mass scan results conducted once a week
throughout the campaign period. In this way some masses changed during the total observation time because of
changed scan intensities and the limited number of masses to be selected. Time resolution was set to 270 s, i.e.
4.5 min. The dataset was averaged after the campaign for 30 min and 1h for comparison with other less time
resolved measurement data. The drift tube pressure (pdrift) was kept between 2.1 and 2.3 mbar with a mean of
2.2 mbar. The detection chamber pressure was kept at $2x10^{-5}$ mbar. The intensity of the reference ion signal for
detection efficiency, i.e. m/z = 21, was recorded as $(4.4\pm1.0)\times10^7$ counts per second. For more details on the set-
up see Bourtsoukidis et al. (2014). A list of all recorded masses can be found in the supporting online
information. Because the PTR-MS technique does not allow for a detailed chemical structure analysis, the
cartridge and canister samples were used as complementary information as to the identity of masses with more
than a single compound present.

**2.2.4    Particle Number Concentration and Surface Area Measurements**
The aerosol inlet was located 3.5 m above ground, about 1 m above the measurement van roof, attached
to an aerosol splitter (Leibniz Institute for Tropospheric Research (TROPOS), "Kuh"). A LVS pump (Leckel
GmbH, Berlin) operated at 1 m³ h⁻¹ corresponding to an aerosol flow of 138 cm³ sec⁻¹ and a PM10-head (Leckel
GmbH, Berlin) suitable for cut of at 10 µm with 2.3 m³ h⁻¹ was used to reduce diffusion losses. This served all
particle measurement instruments.

The instruments that measured particle number (PN) and particle size distribution included a GRIMM

1.108 (particle sizes in optical equivalent diameter, GRIMM Aerosol Technik GmbH & Co. KG, Ainring),
GRIMM 5.403, and GRIMM 5.416 (particle sizes in mobility equivalent diameter). Sampling average was
mostly 1 min and 8 minutes for Grimm 5.403.

The GRIMM 5.416, a condensation particle counter with n-butanol, provided total PN count over a size

range from 4-3000 nm at a flow rate of 1.5 L min⁻¹, and the uncertainty for 1 min sampling was ± 0.1% or ± 15
cm⁻³ (Helsper et al., 2008;Wiedensohler et al., 2017). The GRIMM 5.403, a scanning mobility particle sizer
equipped with a long DMA combined with a CPC with n-butanol measured particle number concentrations with
size distribution information for particles between 10-1100 nm at a sample flow rate of 0.3 L min⁻¹ and a sheath
flow rate of 3 L min⁻¹. For technical details see Heim et al., (2004). The uncertainty associated with the
measurement is size dependent, with an uncertainty range of 10-15% in the lowermost size range and approx. 2-
3% in the upper size range, and a total of 44 size bins. The GRIMM 1.108, a portable laser aerosol spectrometer
and dust monitor measured particle number concentration with size distribution information, covering 350-22500
nm, with a sampling flow rate of 1.5 L min⁻¹. Particle number concentrations were determined for 15 size bins
with an uncertainty of ± 3%. For technical details see Görner et al. (2012).

The TSI Nanoparticle surface area monitor 3550 (NSAM) measured lung deposable surface area for

particle sizes ranging from 10-1000 nm at a flow rate of 2.5 L min⁻¹. These values are reported in units of µm²
cm⁻³ corresponding to empirically derived parameters that correspond to the regions where the particles are
deposited in the lung. Alveolar deposition was measured. Measurement accuracy for the NSAM was ± 20% for
both parameters. Further instrument and measurement details are described elsewhere (Kaminski et al.,
2013;VDI, 2017).

The NSAM was calibrated at the German Environment Agency (UBA, Langen) with instruments from

IUTA, Duisburg (Kaminski, 2011), the GRIMM 1.108 was sent in for maintenance and re-calibrated at the
manufacturer prior to use in the campaign, while all other instruments were calibrated a priori at the TROPOS
aerosol calibration facility in Leipzig (Weinhold, 2014).

A continuous aerosol size distribution (0.01 µm to 30 µm) was created using a combination of GRIMM

5.403 (0.01 µm to 1.1 µm) and GRIMM 1.108 (0.3 µm to 30 µm). Averaged 1-h size distribution from both
particle instruments were merged to create a full size distribution from 0.01 to 30 µm. Size distributions from the
two analyzers were merged by considering GRIMM 5.403 for particles sizes <1.1 µm and sizes equal or above
1.1 µm uses GRIMM 1.108. At 1.1 µm both individual logarithmic size bin boundaries of the 5.403, and 1.108
were most similar allowing "a smooth merge" without losing any size bins. We also assumed that the particles
were spherical and thus no adjustments were made in the size bins, nor were any adjustments made for possible
differences in aerodynamic vs optical derivation of diameter.

**2.2.5   Ceilometer**

State-of-the-art ceilometers provide the vertical profile of aerosol backscatter (Wiegner et al., 2014).

There are numerous approaches to estimate the mixing layer height (MLH) from the measured profile; the
underlying assumption is that at the top of the mixing layer aerosol concentration drastically drops resulting in a
pronounced decrease of backscattered signal intensity. Measurements in the framework of BAERLIN2014 were
performed with a Vaisala ceilometer CL51 (Münkel, 2007;Geiß et al., 2017). This instrument is eye-safe (class
1M), operated fully automated and unattended. The diode laser emits at a wavelength of 910 nm; the absorption
by water vapour can be ignored as long as only the MLH is to be determined (Wiegner and Gasteiger, 2015).
Laser power and window contamination are permanently monitored to ensure long-term stability. Due to the one
lens design the lowest detectable layers are around 50 m, and the system is capable to cover an altitude range
greater than 4000 m, topping out around 8 km. Signals are pre-processed, e.g. for the suppression of noise
generated artefacts. The range resolution is 10 m, and the temporal averaging is 10 min.
The heights of the near surface aerosol layers were analysed by a gradient method from the backscatter
profiles in real-time (Emeis et al., 2008) with a MATLAB-based software which is provided by the manufacturer
and has been improved continuously (Münkel et al., 2011). The minima of the vertical gradient is used to
provide an estimate of the MLH (Emeis et al., 2007). All MLH data presented are following this method (for
more detail see Schäfer et al. (2015)) unless otherwise noted. The influence of different options of the
proprietary software and an comparison with the more sophisticated approach COBOLT (COntinuous BOundary
Layer Tracing) on the retrieved MLH is discussed in detail by Geiß et al. (2017). It was found that the
proprietary software slightly tends to overestimate the MLH compared to COBOLT.
The various instruments outlined above had differing sampling times and so for those instruments that
provided real-time or higher time resolution data, a 30 minute average will be used in the data presented here for
comparability.

### 2.2.6 $PM_{10}$ Filter Analysis

Prior to sampling, the quartz fiber filters were baked at 800°C under synthetic air to remove impurities.
Post-sampling, the $PM_{10}$ filters were analyzed for total mass, elemental carbon (EC), water soluble and total
organic carbon, chloride, sulfate, nitrate, sodium, ammonium, potassium, calcium, and organic tracers.
HYSPLIT back trajectories (based on GDAS meteorological data) were calculated for 72 hours over the time
period of each filter with a new trajectory each 6 hours for air masses ending at ground level (at the monitoring
station) (Stein et al., 2015). Back trajectory plots are included in the Supplemental Information following the
final filter groups. Based on similarities in the bulk composition analysis and HYSPLIT back trajectory
information, the filters were grouped before being extracted and analyzed for organic tracers. Not all filters were
included in these groups, so as to create groups that showed significant similarities. Some individual filters were
therefore also excluded from the organic tracer analysis because of a lack of remaining OC mass.
$PM_{10}$ mass was first quantified gravimetrically and then analyzed for elemental and organic carbon.
For this the filter samples were heated to 750°C in an oxygen stream. The gas stream was then passed through an
oxidation catalyst to ensure complete oxidation of the organic carbon to carbon dioxide ($CO_2$). In contrast to the
organic carbon, elemental carbon is directly oxidized at higher temperatures without the requirement of a
catalyst. The organic carbon, as $CO_2$, was then detected using a cavity ring-down spectrometer (Picarro Inc.).
The distinction between the elemental and organic carbon fractions in the samples was based on the temperature
profile during the analysis. For more details see Ehlers (2013) and Kofahl (2012).
A portion of the filter (1.5 cm²) was water extracted to determine water soluble organic carbon (WSOC)
using a TOC-V SCH Shimadzu total organic carbon analyzer (Miyazaki et al., 2011;Yang et al., 2003). The
remaining amount of OC was calculated as water insoluble organic carbon (WIOC). A fraction of the remaining
solution was used to analyze for water soluble anions and cations by ion chromatography (Dionex ICS 2100 and
Dionex ICS 100) (Wang et al., 2005). For the organic tracer analysis, filters were composited as per the bulk
composition and HYSPLIT determined groups and extracted with 50/50 dichloromethane and acetone by
sonication, an aliquot was derivatized and analyzed by GC-MS (GC-6980, quadropole MS-5973, Agilent
Technology) for organic molecular marker compounds, as described in more detail by Villalobos et al. (2015)
and references therein. Approximately 150 organic tracer species were analyzed for, of which less than 100 had
concentrations regularly above the detection limit. A limited subset of these was then used in the source
apportionment analysis.

**2.3 Chemical Mass Balance for Source Apportionment**
A chemical mass balance analysis of the organic carbon fraction of the $PM_{10}$ filter samples was carried
out using the organic tracer information. Source apportionment analysis using the CMB technique provides an
effective variance least squares solution for a set of linear equations that include the uncertainties of the input
measurements, and have been applied to the mass balance receptor model (Watson et al., 1984). As such, it
allows for the estimation of the contribution of different source categories to the ambient concentrations
measured at any one location, in this case an urban background site in Berlin. The species included in the CMB
analysis were levoglucosan, 17α(H)-21β(H)-30-norhopane, 17α(H)-21β(H)-hopane, benzo(b)fluoranthene,
benzo(k)fluoranthene, benzo(e)pyrene, benzo(a)pyrene, and C27-C33 alkanes. The US EPA CMB Software
version 8.2 was used. Source profiles for vegetative detritus (Rogge et al., 1993), wood burning (Fine et al.,
2004), diesel and gasoline motor vehicles (Lough et al., 2007) were included in the final result. In addition, a
profile for poorly maintained vehicles ('smoking vehicles') (Lough et al., 2007) was evaluated but found
inappropriate. The link between tracers and sources is discussed in further detail in section 3.5.2. The secondary
organic aerosol fraction was calculated based on WSOC not related to biomass burning (Sannigrahi et al., 2006).
The fitting statistics for the final result are shown in Table 3.

**3   Results & Discussion**

**3.1 Time Series and Diurnal Cycle**
The 30 min data time series of $O_3$, $NO_2$, NO, CO, benzene, toluene, and $PM_{10}$, along with basic
meteorological data from the BLUME station in Neukölln and MLH as derived from the proprietary software are
shown in Figure 2, spanning the duration of the campaign. All times are given in CET. The 8 h mean ozone
concentrations show that the EU target value for ozone (120 µg m$^{-3}$ based on 8 h means) was exceeded 6 times
during the measurement period, and the WHO guideline (100 µg m$^{-3}$) was exceeded 18 times. The hourly limit
value for $NO_2$ (200 µg m$^{-3}$) was not exceeded, though concentrations often exceeded 100 µg m$^{-3}$. The daily limit
value for $PM_{10}$ (50 µg m$^{-3}$) was not exceeded.
Elevated concentrations were often observed at the same time for many of the pollutants included in
Figure 2, with the exception of ozone. Ozone, as a secondary pollutant formed photochemically from $NO_x$ and
NMVOC precursors, follows a similar pattern to temperature (Pearson correlation coefficient [standard error] of
0.82 [0.014]), and peaks at different times than the primary pollutants. The formation of ozone can be limited by
either $NO_x$ or NMVOCs, depending on the ambient concentrations which are controlled by sources (e.g.,
vehicles, biogenics) and transport. $NO_2$, NO, CO, toluene, and benzene all have diurnal cycles that peak in the

morning and evening, reflecting their anthropogenic traffic-related emission sources (see Figure S1 in SI). The morning peak in the pollutants occurred at 7 or 8 am, while the evening peak occurred quite late between 9 and 11 pm, likely owing to a combination of daytime emissions and the decrease in the MLH. Traffic counts, from MC143 and MC220 in Neukölln (see location in Figure 1), showed that traffic increased dramatically between 6 and 8-9 am, after which a slow but steady increase led to a peak at 5-6 pm, after which the traffic count dropped dramatically. In contrast, ozone, temperature, and mixing layer height followed parallel diurnal cycles with a minimum at 6 am and a broad afternoon peak between noon and 6 pm. During BAERLIN2014 the maximum height of the mixing layer was found to be 1.5-2 km between noon and 18:00 and below 500 m during the night/early morning. These numbers indicate the vertical extent of the urban pollution layer over the measurement site where pollutants are most likely residing. Relative humidity showed the opposite with a peak at 6 am, and a broad low between noon and 6 pm.

These results are supported by the Pearson correlation coefficients among $NO_2$, $NO$, $CO$, toluene, and benzene, which for hourly values range from 0.51-0.82 (all statistically significant at an alpha=0.05; see Table S2), with the strongest relationship between $CO$ and $NO_2$. The correlation to relative humidity was found to be negative for MLH (-0.66 [0.022]), temperature (-0.71 [0.014]), and ozone (-0.76 [0.014]). The pollutant with the strongest relationship to temperature was ozone.

The time series of particulate matter mass ($PM_{10}$), derived $PM_1$, $PM_{2.5}$, and $PM_{10}$ mass from the GRIMM 1.108 particle number size distribution measurements, total particle number, and particle surface area are shown in Figure 3. While the two $PM_{10}$ time series along with the PM and particle number time series associated to the same instrument (GRIMM 1.108) are most similar, the other total particle number time series do not show significant similarities. This is largely owing to the difference in size fractions measured by the different instruments. Correlation analysis of the pollutant concentrations from Neukölln with MLH values on the basis of averaged diurnal cycles of hourly-mean values (in our case monthly averages during July and August) provided highest correlations with PN for accumulation mode particles (size range 100 – 500 nm) and significant correlations for $PM_{2.5}$ and $PM_1$ (Schäfer et al., 2015) showing similarities to investigations in Augsburg, Germany (Schäfer et al., 2016) and Beijing, China (Tang et al., 2016). In addition to this investigation for the reference site, a more detailed correlation analysis of the MLH with $PM_{10}$, $O_3$, and $NO_x$ taking into account all 16 BLUME stations in Berlin was carried out using the MATLAB approach outlined here, as well as an alternative approach, COBOLT (Geiß et al., 2017). In this context it was assumed that the MLH derived for the reference site in Neukölln is representative for the entire metropolitan area of Berlin. The correlation analysis of the diurnal cycles (averaged over the duration of ceilometer measurements from BAERLIN2014) of the MLH and $PM_{10}$ found that correlations were completely different at the different sites regardless of site type, indicating that surface concentrations of $PM_{10}$ were not predominantly determined by the MLH, but rather by local sources and sinks, and meteorological factors, among others. In the case of $O_3$, strong positive correlations were identified for both the BLUME sites on the periphery of Berlin, as well as the urban background locations. In contrast, for $NO_2$, a negative correlation to MLH was observed for all sites at the periphery of the city, and to a lesser extent at some of the urban background sites (Geiß et al., 2017).

Particle size distribution during the study period is shown in Figure 4. Size distribution was dominated by ultrafine number size distribution ("UFP", <100 nm) throughout the day (i.e. particle formation close by). The number and volume distribution was further binned into at least 5 size bins, as presented in Figure 4 for comparison with other urban background measurements. The average daytime total number and volume

concentration remained in the range of 5.5 - 6.0 x $10^3$ cm$^{-3}$ and 11 - 12 µm$^3$ cm$^{-3}$, respectively, in contrast to the
stronger signal during the nighttime. The mean (median) total number and volume concentration over the entire
measurement period was 6.1 x $10^3$ cm$^{-3}$ (5.4 x $10^3$ cm$^{-3}$) and 11.8 µm$^3$ cm$^{-3}$ (9.5 µm$^3$ cm$^{-3}$), respectively. Over
80% of the total number concentration is ultrafine particles, and the contribution is higher during the nighttime.
Volume distribution is largely dominated by the accumulation mode particles which is typical of many urban
sites. The number concentrations were similar to other urban stations in Germany (Birmili et al., 2016).
The diurnal cycles for total PN for the three instruments covering the smaller particles (excluding the
observations from the GRIMM-1.108) have morning and evening peaks, similar to the diurnal cycle for NO$_2$,
indicating a traffic origin. The diurnal cycle for the larger particles, as sampled by the GRIMM-1.108 has a much
more dominant early morning peak and mid-afternoon minimum, without the second evening peak.
In Figure 5, at least two major contributors to UFP over the course of the day could be identified, in the
morning and during the night. The presence of the morning peak is likely due to traffic-related emissions. Such a
peak has also been identified in other species, as well as other studies in urban areas (Borsós et al.,
2012;Mølgaard et al., 2013). There was a gradual increase in the UFP concentration from late afternoon which
continues overnight till early morning hours. This nighttime feature of UFP was observed during weekends as
well as on the weekdays. The reasons for this could be that the source contributing to this is something other than
or in addition to traffic and may be active or enhanced overnight, the decrease in mixing layer height at night
traps the particles in a smaller volume compared to daytime, and/or that night time deposition of particles is
lower than daytime owing to higher atmospheric stability. The co-located trace gas measurement showed that the
elevated UFP nighttime concentration correlates with toluene, among other gases such as CO.  Daily
observations also showed occasional and episodic "particle burst" (new particle formation) events for particles in
the size range of 10-50 nm, which could be related to fresh plumes or to regional particle formation events.

**3.2  NMVOC measurements – Method comparison**
The results of the four NMVOC measurement methods were compared and contrasted for benzene and
toluene. While differences in e.g., instrumentation and measurement technique (mass-to-charge (m/z) ratios vs
compounds), inlet location, and time resolution, do not allow for direct comparisons, a comparison can be useful
to understand how different or similar the information provided by the various methods can be. A summary of
these methods and the compounds measured, including information on the detection limits and sampling times is
provided in Table S1.
The 30-min data reported from the BLUME city air quality monitoring network was compared to the
PTR-MS data for m/z 79 (benzene) and m/z 93 (toluene), as both instruments provide high time-resolution data.
The correlations between the two methods were good given the imperfect nature of the comparison, both with
Pearson's r values for benzene and toluene of 0.39, significant at the p<0.05 level. The lower correlation values
were likely owing to a number of factors including the differences in measurement method, and in location of the
inlets for each instrument and thereby source influences – one of which (PTR-MS) was located on the street side
of the van at approx. 2.5 m above ground, while the other (BLUME) was located above the measurement
container approx. 5 m from the street. The inlet at the street would be influenced more directly by vehicle
emissions in comparison to the inlet above the measurement container, which is especially relevant in that the
PTR-MS was likely influence by individual vehicles, while this would not be the case for the container
measurements. This influence of vehicles on the PTRMS data at higher time resolution is supported by an
increase in Pearson's r values with longer averaging times, which reduces the influence of individual vehicles.
For 1 h (3 h) average concentrations the r values increase to 0.48 (0.58) and 0.53 (0.71) for benzene and toluene,
respectively, all significant at the $p < 0.05$ level. Furthermore, the Pearson's r values for the correlations between
the BLUME network and the individual canisters were 0.39 (benzene) and 0.83 (toluene), both statistically
significant with p-values $< 0.05$, and between BLUME and the cartridge samples 0.51 (toluene) and not
significant for benzene. All benzene and toluene measurements are shown in Figure S2.
In order to investigate the possibility of identifying molecular structures of PTR-MS derived m/z
measurements, a comparison of the continuous measurements of the PTR-MS and intermittent canister samples
was also carried out. For a number of cases only one compound quantified from the canister samples matched a
specific m/z, while in other cases multiple compounds were quantified in the canister samples that had the same
mass. For example, propanal, acetone, n-butane, and 2-methylpropane all have a molecular weight
corresponding to m/z 59 (molar weight $M_W = 58$ g/mole + $M_W(H^+) = 1$ g/mole), among which the PTR-MS
cannot distinguish. In some cases, the fractional contribution of compounds with the same m/z ratio was
relatively similar across all canister samples, as for o-xylene, m+p-xylene, and ethylbenzene (m/z 107). However
this was rather the exception, with relative contributions more typically showing significant variation among the
canister samples (see Figure S3 in the SI). Correlations between the canister samples and PTR-MS results were
carried out for 35 individual m/z values for which at least one compound was quantified in the canister samples.
While the absolute r values of the correlations ranged from 0.00016 to 0.63, the correlations were generally quite
poor, showing little to no correlation for many of the m/z (only 9 of the 35 total number of m/z values evaluated
had r values greater than 0.3), with no systematic bias identified. There are a number of reasons for this, beyond
the difference in how the instruments measure (m/z vs compounds), such as inlet location and sampling time.
Previously, in a targeted inter-comparison experiment where whole air samples (canisters) were compared with
online PTR-MS measurements, differences of as little as 20 s in the sampling intervals contributed to scatter in
the comparison of the two measurements that was especially relevant for the more reactive NMVOCs (de Gouw
and Warneke, 2006). Additionally, scatter in inter-comparisons between ground-based fast time response and
GC-MS systems was found to be typical (Lerner et al., 2017) and references therein). In the context of this study,
the measurements should not be considered as an inter-comparison since, as described above, the inlets were
approx. 5 meters apart, at different heights above ground level, with one street-side and the other above a
measurement container. For these reasons, while both measurements are valid, as this comparison shows, the
differences in quantification method, but also importantly instrument location and set-up result in substantial
differences in what is being quantified so that the comparison is limited in value.

**3.3 NMVOC Measurements – Characterization of different locations by canister sampling**
The average fractional contribution to mixing ratio by compound class for each of the Neukölln,
Altlandsberg, Plänterwald sites, the Tiergarten tunnel and the AVUS motorway samples is presented in Figure 6.
The number of compounds included in each class was: alkanes (19), alkenes and alkynes (13), aromatics (14),
oxygenated (6), and biogenics and their oxidation products (5; referred to as 'biogenics' for simplicity). For a
complete list of the compounds and their grouping, see the supplemental information. In the following text and
figures two extremely high values for acetone were removed (one sample from the Neukölln station, and one
from the Altlandsberg samples). Since these two values were extreme outliers, their origin remains unclear.
Therefore we have removed them from the averages and treated them separately. (Text is included in the SI to
demonstrate how these two values change the results presented here.) The largest contributions of the quantified
VOCs to mixing ratio were from the alkanes (27 - 41 %) and oxygenated (23 - 55 %) compounds. Biogenics
were always a minor contribution to mixing ratio, but their contribution was largest in the Plänterwald samples
(11 %) and negligible at the two traffic locations. Alkenes/alkynes and aromatics showed the largest contribution
to mixing ratio at the traffic sites, at 17 - 23 % and 14 %, respectively. The highest total NMVOC mixing ratio of
those compounds measured here was found at the traffic sites (Tiergarten tunnel, 64 ± 17 ppbv; AVUS
motorway, 170 ± 82 ppbv; average mixing ratio ± standard deviation among the samples). The total mixing
ratios of the 57 measured compounds at Altlandsberg and at the urban background station in Neukölln, showed
similar results, with an average mixing ratio and standard deviation of 14 ± 6.4 ppbv and 19 ± 5.6 ppbv,
respectively. The mixing ratios found in Plänterwald were similar to the urban background location, with an
average of 17 ± 3.4 ppbv, although with a larger contribution from biogenics. In comparison, total NMHC
mixing ratios for urban background in Paris during the MEGAPOLI winter campaign was 12 ppbv (midnight
median levels) or 17 ppbv (maximum of median daily values), with somewhat lower mixing ratios measured
during the summer campaign (Dolgorouky et al., 2012;Ait-Helal et al., 2014).
Previously, a measurement campaign was carried out during June-August of 1996 in Berlin, during
which samples were taken at the Neukölln urban background station, as well as at a traffic station on Frankfurter
Allee. During this campaign, VOC measurements were taken 4 times a day for 2 hours over the course of one
week (7 days) of each month using bag samples, adsorption tubes and DNPH cartridges and analyzed by gas-
chromatography (Thijsse et al., 1999). This provides a good basis for comparison to the NMVOCs measured by
canister sampling (most similar in method) during this campaign almost 20 years later. Overall, the mixing ratios
for most compounds that were measured in both projects at the urban background location in Neukölln were
lower now than in 1996 (Figure 7). For the traffic locations the results are less clear. Given that the Frankfurter
Allee monitoring station is a traffic station, these measurements would likely be more comparable to the
Tiergarten Tunnel measurements of this study, rather than those samples taken during a traffic jam on the AVUS
motorway where concentrations were extremely elevated. Indeed, the mixing ratios measured during the traffic
jam were found to be higher in most cases than those measured in 1996 at Frankfurter Allee. However, the
comparison between the Tiergarten Tunnel measurements and Frankfurter Allee showed much more similar
results to those of the urban background station comparison, with concentrations generally being lower today
than approx. 20 years ago (Thijsse et al., 1999).
There are a couple of exceptions in this comparison, where the mixing ratios measured in this campaign
stand out as substantially higher than those measured 20 years ago. Considering only those few compounds that
have a ratio of 0.6 or less for the average mixing ratio in 1996 relative to that in 2014, the biogenic contributions
in Neukölln (isoprene (0.3), methylvinylketone (0.1)) show increases. These increases may be attributable to
changes in vegetation around the measurement site. Other NMVOCs, such as cis-2-butene and cyclopentane
showed increases for both the urban background site and traffic site (Tiergarten Tunnel vs Frankfurter Allee).
Other compounds, such as cis-2-pentene and trans-2-butene (traffic site) and 1,2,3-trimethylbenzene (urban
background) showed increases at only the one site type. While the literature on trends of NMVOCs is limited,
data from a traffic site in London, a rural background site in the UK, and a remote site in Germany showed that
over the period from 1998-2009 all individual NMVOCs evaluated (with the exception of n-heptane at the rural
background site) were decreasing, with stronger decreases observed at the traffic site relative to the other site
types (von Schneidemesser et al., 2010). Similarly, an evaluation of C2-C8 hydrocarbon data, as total HCs and
by compound class, for a number of sites across the UK from 1994-2012, also documented decreases across all
compound classes (Derwent et al., 2014). Finally, a broader evaluation of the trends in anthropogenic NMVOC
emissions across Europe also documented a decrease between 2003 and 2012 (EEA, 2014, 2016). As such, the
existing literature does not provide any detailed documentation that might be able to address the potential
increases in those few compounds here where an increase was observed. Furthermore, longer-term sampling may
show that the increases documented here do not reflect the long-term trend.
**3.4 OH Reactivity**

To better understand the role of these compounds with respect to their role in ozone formation and the
reactivity of the measured compounds, the reactivity with respect to OH ($R_{OH}$) was calculated. These results are
shown in Figure 6 and parallel the results presented for the mixing ratios. In all cases, including other studies
discussed, the values presented are calculated OH reactivity based on measurements of NMVOCs and not OH
reactivity that was measured directly. Because the OH reactivity estimates are based on a limited number of
NMVOCs, the values presented here are a lower limit. The relative importance of the biogenics, alkenes and
alkynes, and to a lesser extent the aromatics increased when considering OH reactivity as is visible in Figure 6
(for a complete list of compounds included in these classes, see the SI). The largest contribution to OH reactivity
was from either the biogenics and their oxidation products (0-75%) or the alkenes and alkynes (10-55%),
depending on the location, with the alkenes and alkynes dominating at the traffic locations, where the biogenic
contribution was negligible. The NMVOCs included in each of these categories are provided in Section S1. The
contribution to OH reactivity from alkanes ranged from 4% (Plänterwald) to 18% (AVUS motorway). The
contribution from oxygenated compounds, despite their substantial contribution to mixing ratio, ranged from
only 5-13% of OH reactivity. That said, only 6 oxygenated NMVOCs (of 57 total NMVOCs) were included
here, and a recent study by Karl et al., (2018) found an appreciably greater fraction of oxygenated NMVOCs in
urban areas than previous studies identified. The molar flux of oxygenated NMVOCs being actively emitted into
the urban atmosphere from measurements in Europe was found to be $56 \pm 10\%$ relative to the total NMVOC flux
(Karl et al., 2018), which indicates that a much larger contribution from oxygenated NMVOCs is possible if
different measurement techniques are used. The contribution to the biogenic OH reactivity at Plänterwald
originated largely from isoprene (88%), with 7% from α- and β-pinene. Similar contributions were found at
Neukölln and Altlandsberg. The mean (median [$25^{th}$, $75^{th}$ percentile]) total OH reactivity from the 57 species
was 2.6 s$^{-1}$ (2.6 [2.1, 3.0] s$^{-1}$) at Neukölln, and ranged from 2.2 s$^{-1}$ (2.2 [1.5, 2.8] s$^{-1}$) at Altlandsberg to 34 s$^{-1}$ (34
[29, 39] s$^{-1}$) from the AVUS motorway. While studies have shown that a number of NMVOCs, such as isoprene,
or other terpenes can also have anthropogenic sources (Derwent et al., 2007;Reimann et al., 2000), we treat them
as biogenic and do not try to tease apart the biogenic vs potential anthropogenic contributions in this context.
An earlier study (BERLIOZ) also made measurements of $C_2$-$C_{12}$ NMHCs in Berlin and at sites in the
surrounding area, mostly focused on the production of ozone in downwind locations of the city (Winkler et al.,
2002;Volz-Thomas et al., 2003;Becker et al., 2002). They report OH reactivity for two sites outside of Berlin,
Blossin (approx. 15-20 km southeast of the Berlin city boundary) and Pabsthum (approx. 30-35 km northwest of
the Berlin city boundary). The total OH reactivity reported at these sites range between 1 - 7 sec$^{-1}$ and approx.
0.25 - 2 sec$^{-1}$, respectively. These are similar to those values found at the urban background locations in Berlin,
with the most comparable location being Altlandsberg (2.2 s$^{-1}$). The contribution from isoprene to the OH
reactivity was found to be 70% at Blossin and 51% at Pabstthum, on average, although during the passing of a
city plume at Pabsthum 46% of reactivity was contributed by isoprene, with the remaining contribution
attributed to anthropogenic NMHCs (Winkler et al., 2002).

The total OH reactivity values of measured VOCs in Berlin (2.6 s$^{-1}$) are similar to the average total OH

reactivity from VOCs observed in other European cities, such as Paris (approx. 4.0 s$^{-1}$) and London (1.8 s$^{-1}$)
(Dolgorouky et al., 2012;Whalley et al., 2016), and, not surprisingly, lower than those observed at cities in the
Pearl River Delta region of China (8-14 s$^{-1}$). Specifically, Liu et al. (2008) reported OH reactivity from a
measurement campaign in Ghangzhou and Xinken during one month in the autumn of 2004. The OH reactivity
from alkanes, alkenes, and aromatics from Ghangzhou was reported to be 1.9 ± 1.5 s$^{-1}$, 8.8 ± 6.8 s$^{-1}$, and 2.9 ±
2.7 s$^{-1}$, respectively. In all cases, these values are about one order of magnitude greater than those calculated for
the urban background locations during this campaign (see Table 2). The level for isoprene (0.5 ± 0.4 s$^{-1}$)
however, was much more similar to the OH reactivity reported for the biogenics at the urban background
locations in this study. In London, OH reactivity of alkanes, alkenes+alkynes, aromatics, and biogenics was
reported to be 0.81 s$^{-1}$, 0.47 s$^{-1}$, 0.235 s$^{-1}$, and 0.25 s$^{-1}$, respectively, which are values much more similar to those
in this study (Whalley et al., 2016). The relative importance of alkanes and alkenes+alkynes was the reverse for
London compared to Berlin.

In the MEGAPOLI winter campaign in Paris, total calculated mean OH reactivity was reported to be

17.5 s$^{-1}$, although this included not only NMVOCs, but also methane, CO, NO, and NO$_2$ (Dolgorouky et al.,
2012). The OH reactivity attributed to the 29 non-methane hydrocarbons and oxygenated VOCs was 23% (4.0 s$^{-1}$
) of the total, somewhat higher than those values reported here (57 NMVOCs) for the urban background
locations. Comparing to the OH reactivity values in Berlin is difficult, since for the winter campaign in Paris,
Ait-Helal et al. (2014) report that the concentrations of the VOCs are generally shown to be lower during
summer, specifically for many of the anthropogenic compounds, although this does vary by compound.
Therefore, the OH reactivity values for Paris considered here should be considered an upper limit for the
comparison with this study. The calculated mean OH reactivity attributed to NO and CO was 1.75 s$^{-1}$ each, and
9.63 s$^{-1}$ for NO$_2$ in Paris (Dolgorouky et al., 2012). By comparison, the mean OH reactivity calculated for
August (to match the time during which the canister samples were taken at Neukölln) was 0.58 ± 1.2 s$^{-1}$ and 0.87
± 0.30 s$^{-1}$ for NO and CO, respectively, and 4.5 ± 3.0 s$^{-1}$ for NO$_2$, which is again, lower, as with the VOCs, but
not unreasonable given the context of the comparison.

Finally, while the 57 NMVOCs included here to calculate OH reactivity were chosen to facilitate

comparison to previous studies, a more exhaustive list could change the picture. For example, as mentioned
above, the limited number of oxygenated NMVOCs measured would likely lessen the contributions of the other
compound classes. As an example, adding six additional oxygenated NMVOCs (propanal, 2-butanol, 1-propanol,
butanal, 1-butanol, pentanal) increased the total average OH reactivity between 0.12 s$^{-1}$ (Plänterwald) to 1.7 s$^{-1}$
(AVUS Motorway). The percent contribution of these six oxygenated NMVOCs ranges between 2.5% and 9.3%
of the new total OH reactivity. In contrast, a similar analysis that included three additional biogenic NMVOCs
(limonene, sabinene, eucalyptol) showed much smaller additional reactivity, never more than 0.02 s$^{-1}$. These
compounds also were not consistently present across all samples.

**3.4.1    OH reactivity – direct comparison to a previous study in London and Paris**

As a comparison to the R$_{OH}$ estimates calculated for London and Paris based on approx. 10 years of

monitoring data through 2009 (von Schneidemesser et al., 2011), a subset of the NMVOCs was taken to enable a

more equal comparison to the values reported for summer (JJA) in that study. The only difference in the compounds included is the contribution of n-butane, which was not included in the Berlin calculations because of a local source of contamination (in London the contribution of n-butane to OH reactivity from this subset of NMVOCs was approx. 5% or less). The referenced study was focused on the contribution of biogenics, specifically isoprene, to OH reactivity. At the London Eltham site (urban background) isoprene contributed 25% to the OH reactivity for summer and 16% at Paris Les Halles, also an urban background location (24 total NMVOCs, including 9 alkanes, 9 alkenes/alkynes, 5 aromatics, 0 oxygenated, 1 biogenic) (von Schneidemesser et al., 2011). Using the reduced, matched set of compounds, isoprene accounts for 37% of OH reactivity at the Neukölln location on average, and as much as 82% at the Plänterwald (urban park) location in Berlin. The Neukölln urban background location values are a bit higher than those in London and Paris, although not dramatically different. The Plänterwald urban park location however, demonstrates the importance of such areas for the biogenic influence on OH reactivity, especially considering that even at Harwell, a rural background location west of London in the UK, isoprene contributes on average only 10% of OH reactivity. Although, as pointed out in the study, this is likely an underestimation of the biogenic importance given that only isoprene is included and for northerly regions other biogenics, such as monoterpenes may play a more important role (von Schneidemesser et al., 2011).

### 3.5  $PM_{10}$ Filters

#### 3.5.1    Bulk composition and HYSPLIT back trajectories

The $PM_{10}$ filters were analyzed for water soluble and water insoluble OC, EC, and ions. In addition, filter samples were grouped to ensure enough mass for analysis of organic molecular markers. The groups were informed by the bulk composition analysis results, including the ratio of water soluble to total OC and the ratio of ions to OC, and HYSPLIT back trajectories. Back trajectories were evaluated to provide information on the origin of the air masses and source-receptor relationships (Stein et al., 2015). The results of this bulk composition analysis are shown in Figure 8. Select individual filters that had sufficient mass and did not fit with any of the other groups were analyzed individually (B17, B19, B30). All values listed for groups are an average of the results from the filters included in the group. The air mass origins as per HYSPLIT are summarized in Table 3 (see also Figure S4).

Groups A, B, C, and D show significant similarity in their percent of OC that is WSOC, which ranges from 27 to 34%. The ratio of ions (sulfate, nitrate, ammonium) to OC is however, very different. Groups B and C have an ions:OC ratio of 1.2 and 0.98, while groups A and D have ratios of 0.56 and 0.50, respectively. The $PM_{10}$ mass loadings for B (20 µg m$^{-3}$) and C (24 µg m$^{-3}$) were lower than for A (27 µg m$^{-3}$) and D (35 µg m$^{-3}$), see Table 3. The concentrations of EC ranged between 1.1 and 1.9 µg m$^{-3}$ but did not group as with the other species, with the lowest concentration in group B and the highest in group C.

Group E had a very low percent of WSOC (19%) and an ions:OC ratio of 0.59. It also had the lowest $PM_{10}$ mass (20 µg m$^{-3}$), and either the lowest or among the lowest concentrations for all ions. The OC concentration however, was 5.5 µg m$^{-3}$, which was roughly in the middle of the OC concentrations measured, while the EC concentration was also the lowest at 0.71 µg m$^{-3}$.

B17, B19, and B30 were analyzed individually because their bulk composition analysis and back trajectory patterns did not group well with the others, and sufficient mass was available for tracer analysis without needing to composite filters (Table 3, Figure 8). B17 and B30 had a higher percent WSOC (66% and

56%, respectively), and ions:OC ratios of 1.3 and 2.4, respectively. 37% of OC was WSOC for B19, and the
ions:OC ratio was 0.77. Total $PM_{10}$ mass was 38.8 µg m$^{-3}$, 31.0 µg m$^{-3}$, and 39.5 µg m$^{-3}$, and OC concentrations
were 7.0 µg m$^{-3}$, 5.9 µg m$^{-3}$, and 3.9 µg m$^{-3}$, for B17, B19, and B30, respectively. All three samples had
significantly larger contributions from sulfate, and to a lesser extent also higher ammonium, compared to the
other groups. B30 also has a large amount of nitrate in contrast to all other samples, and somewhat higher
concentrations of potassium and sodium as well. B17 had the highest concentration of EC (2.3 µg m$^{-3}$) of all
samples.

There were significant concentrations of sulfate across all samples, ranging from 1.2-6.0 µg m$^{-3}$, but

particularly so in B17, B19, and B30. Sulfate is typically attributed to industrial sources, as the content of sulfate
in fuels has been reduced significantly and is now quite low (Villalobos et al., 2015). Sea-salt is in this case not
likely as a source, as Berlin is not within close proximity of a coastal region where such components are
typically identified (Putaud et al., 2004). In general the significant contributions of sulfate, nitrate, and
ammonium are indicative of a secondary inorganic aerosol (ammonium sulfate and ammonium nitrate) (Putaud
et al., 2004;Schauer et al., 1996). Previous work has shown that secondary inorganic aerosol over northwestern
Europe, including Germany, contribute significantly – about 50% – to the $PM_{10}$ concentrations (Banzhaf et al.,
2013). Two studies by Putaud et al. (Putaud et al., 2004;Putaud et al., 2010) summarize the relative contribution
of major constituent chemical species to PM mass, including for near-city and urban background locations. In
comparison to the numbers cited in that study (2004 all European sites; 2010 north-western European sites), the
percent contribution of nitrate (15%; 14%), ammonium (7%; not listed), and sulfate (13%; 14%) to $PM_{10}$ mass at
the urban background site in Berlin were quite similar, ranging from 1-11% (nitrate), 1-5% (ammonium), and 6-
16% (sulfate) in Berlin.

The back trajectories (Figure S4) show that prior to arriving in Berlin, the air masses primarily passed

over Germany for group A. While some additional filters fit the general patterns outlined here, the number of
filters included in the group was reduced to focus more on back trajectories in the group that originated from
over Germany itself. The air masses that characterize group D originated from the Northeast, passing over the
Baltic coast and Poland before arriving in Berlin. For group B the air masses originated from the West over the
Atlantic (not further than 20 degrees W) and passed over northern France, the BeNeLux region and central
Germany before arriving in Berlin. For group C, the air masses originated from the North West, over the North
Sea as far as Iceland, passing between the UK and the Scandinavian Peninsula before arriving in Berlin. Both B
and C had higher concentrations of sodium and nitrate than A and D, while A and D had higher concentrations of
OC and marginally higher concentrations of sulfate than B and C (Figure 8). The air masses of Group E
originated from the North, passing over Scandinavia, the North Sea, or the UK before arriving in Berlin. The
back trajectories associated with B17 and B19 both passed over Poland before arriving in Berlin, with the air
masses associated with B19 extending more northward as well. For B30 the air originates from the West with
some passing over northern France, but mostly comes from over Germany itself. The significant presence of
ammonium and sulfate likely indicates influence of agriculture, as ammonium sulfate is commonly used in
fertilizer and more than 95% of $NH_3$ emissions in Europe originate from agriculture (Harrison and Webb,
2001;Backes et al., 2016;EEA, 2016).

**3.5.2   Organic molecular markers**

The concentrations by composited sample are shown in Figure 9 for the organic molecular markers. Levoglucosan has been established as a molecular marker for biomass burning (Simoneit et al., 1999). The concentrations measured here ranged from 15-60 ng m$^{-3}$. While high concentrations of levoglucosan in urban areas are often associated with residential wood combustion during colder months, it can also be owing to crop burning, wild fires, coal combustion and/or long-range transport of smoke from biomass burning (Simoneit, 2002;Zhang et al., 2008;Shen et al., 2016). The concentrations measured during this summer campaign in Berlin were similar to those measured in PM$_{10}$ from other European cities during summertime, and approx. an order of magnitude lower than concentrations observed in winter (Caseiro and Oliveira (2012) and references therein). The study by Caseiro and Oliveira (2012) confirms the likelihood of agricultural residue burning and/or wildfires as a summertime source for levoglucosan.

Alkanes are useful tracers to distinguish between fossil fuel sources and vegetative detritus. This distinction is informed by the odd-even carbon number predominance, specifically of the $C_{29}$, $C_{31}$, and $C_{33}$ $n$-alkanes to indicate plant material as a source (Rogge et al., 1993). As is visible in Figure 9, the concentrations of those odd $n$-alkanes are much greater than the corresponding even $n$-alkanes. Furthermore, the carbon preference index (CPI) was calculated for the samples using the $C_{29}$-$C_{33}$ $n$-alkanes and ranged from 1.9-5.5, with an average of 3.6. CPI values of approx. 1 are indicative of fossil fuel emission sources, whereas values of approx. 2 or greater are indicative of biogenic detritus (Simoneit, 1986), as is clearly the case for these samples.

Hopanes have been established as markers for diesel and gasoline vehicle emissions, stemming from petroleum product utilization and lubricating oil used in vehicles (Schauer et al., 1996;Rushdi et al., 2006;Simoneit, 1984). The concentrations of the two hopanes measured here and included in the CMB analysis ranged from 0.04-0.13 ng m$^{-3}$ as shown in Figure 9.

Polycyclic aromatic hydrocarbons (PAHs) are formed and emitted most typically during the incomplete combustion of fossil fuels or wood (Ravindra et al., 2008). The concentrations measured during this study ranged from 0-0.23 ng m$^{-3}$ for the individual PAHs shown in Figure 9. These concentrations are similar to, although on the lower end, of those measured in a study in Flanders, Belgium, including measurements at urban locations (Ravindra et al., 2006). Generally, PAH concentrations are lower in summertime owing to lower emissions and shorter lifetimes. The measurements here were conducted during summer, while the measurements in the study in Flanders covered more seasons. To distinguish between sources, PAH concentration profiles or ratios are used. For example, a ratio of benzo(b)fluoranthene to benzo(k)fluoranthene of greater than 0.5 has been identified as an indicator for diesel emissions sources (Park et al., 2002;Ravindra et al., 2008). In this study the ratio ranged from 1.9 to 7.2, indicating a strong influence of diesel emissions for these compounds.

### 3.5.3    Chemical Mass Balance

The molecular markers analyzed in the organic carbon fraction of the PM$_{10}$ samples were used to conduct source apportionment analysis using chemical mass balance. The total OC for these samples ranged from 2.99 to 7.21 μg m$^{-3}$. The amount of OC mass apportioned in the CMB analysis ranged from 21% to 49%. The source profiles included in the model to which OC was attributed includes vegetative detritus, diesel emissions, gasoline vehicle emissions, and wood burning. In addition, a fraction of the unapportioned OC was attributed to secondary organic aerosol based on the unapportioned fraction of water soluble OC and the amount attributed to wood burning, following Sannigrahi et al. (2006). The source contributions to OC, as well as the fitting statistics are listed in Table 4, and shown in Figure 10.

For B17, B19, and B30 the SOA fraction is higher than for any of the others, at 63%, 34%, and 49% of
OC, respectively. They also had the highest concentrations of levoglucosan, ranging from 37.8 to 60.1 ng m$^{-3}$. As
the primary tracer for biomass burning, these three samples also had the largest concentrations attributed to this
source, ranging from 0.22 to 0.44 µg m$^{-3}$ of OC, but the relative contribution was only larger for B30 at 11%. All
other samples had contributions that ranged between 2% and 4% of OC. These three samples had air masses that
originated over Poland (B17, B19) and Germany (B30), indicating a more local-regional source for the biomass
burning. The higher concentrations of potassium in these samples, also an indicator for biomass burning
(Andreae, 1983), provides additional confirmation. The relatively high concentrations of ammonium and sulfate
in these samples may indicate an agricultural influence. Those samples originating from regions to the
West/North had somewhat lower concentrations overall relative to those originating from regions to the
East/North, as shown in Figure 10.
The contribution of diesel emissions ranged from 0.24 - 0.81 µg m$^{-3}$, corresponding to 4 - 21% of OC
fraction. The highest fractional contribution was found in GRC (concentration 0.74 µg m$^{-3}$) (air masses
originating over the North Sea), while the highest concentration was found in sample B17 (fractional
contribution 12%) (from Poland to the East). The diesel from GRC could also have its origin in shipping
emissions, as well as diesel vehicles. High contributions of diesel did not necessarily correspond to high
contributions of gasoline vehicle emissions, which were lower than the contributions from diesel and ranged
from 0.11 - 0.28 µg m$^{-3}$ and 2 - 7% of OC. The highest contribution in terms of fractional contribution and
concentration was found in B30. Furthermore, it should be noted that the source profiles reflect primary organic
aerosol emissions, and therefore the secondary aerosol produced from these vehicular sources, which has been
shown to be substantial in many cases, depending on the control technologies in use (Gordon et al.,
2014a;Gordon et al., 2014b), is not reflected in these attributions.
The contribution of vegetative detritus was among the largest source contributions and ranged from 0.51
- 1.4 µg m$^{-3}$ (11-20%). The relative importance of this source is reflected in the concentrations of the alkanes, as
shown in Figure 9, and their average CPI of 3.6. The largest contribution was found for GRD with air masses
originating over the North Sea.
For all samples, a significant amount of secondary organic aerosol was calculated, 0.87 - 4.4 µg m$^{-3}$ (18
- 63%). While this was the contribution to OC, high concentrations of secondary inorganics (sulfate, ammonium,
nitrate) support the aging of the air masses and the potential for a significant contribution from secondary aerosol
overall.
It should be noted that ambient air samples include contributions from both local sources as well as
emissions that have been transported from locations further away. While the back trajectory analysis is more
relevant for interpreting the influence of emissions from the surrounding region, a comparison to the Berlin
emission inventory reflects on the influence of local source contributions. Both play a role, but neither capture
the complete picture, with limitations in both cases, as discussed further below.

**3.5.4    Source apportionment – emission inventory comparison**
The source apportionment results were compared to the emissions inventory (EI) from TNO-MACC III
(Kuenen et al., 2014). The grid cells for Berlin were extracted and the percent of total emissions for OC by
source category for the Berlin area for June, July, and August as a rough comparison to the source apportionment
results was calculated. Both diesel and gasoline vehicle exhaust sources have significant contributions, although
diesel contributes approx. 19% to total OC emissions in the inventory, whereas gasoline vehicles contribute only
about 1%. Biogenic sources are not included in the inventory. If we focus on the primary sources from the source
apportionment results, the diesel and gasoline vehicles contribute a significant fraction, with diesel comprising a
larger fraction than gasoline vehicles, as in the inventory. The inventory also includes significant contributions
from road transport originating from road, brake, and tire wear, which are not reflected in the CMB results,
owing to the profiles used. About 8% of OC emissions are attributed to agriculture in the EI. This could
contribute to both the biomass burning and vegetative detritus sources; the presence of significant secondary
ammonium and nitrate also indicates an agricultural influence, even though this does not show up in the OC
CMB. In all cases, these primary sources will contribute to secondary inorganic and organic aerosol formation.
The contributions from non-industrial combustion and energy and other industries are not captured as primary
source contributions in the CMB model. Overall, the comparison between the source apportionment results and
the EI is a non-ideal comparison given the differences in methodology and the difference in terms of primary vs
secondary sources that are or are not included. More specifically, the EI provides primary emissions estimates
for a year for all Berlin grid cells (Kuenen et al., 2014), while the CMB results provide source attribution to
ambient concentrations including primary and secondary sources for 3 months of summer at one location in
Berlin. However, one would expect that general patterns are captured for significant sources, as it was for
vehicle emissions, and the indication of agriculture.
**4    Conclusions**
The data presented here provide an overview of the stationary measurements conducted during the
BAERLIN2014 campaign. Of the three main aims of the campaign, two were addressed here, including (1)
characterization of gaseous and particulate pollution, including source attribution, in the Berlin-Potsdam area, (2)
quantification of the role of natural sources, especially vegetation, in determining levels of gaseous pollutants
such as ozone. $PM_{10}$ concentrations and the contributions from inorganic species, such as nitrate, sulfate, and
ammonium that contribute substantially (10-24%) to secondary aerosol were found to be similar in terms of their
relative contribution to $PM_{10}$ in other European cities. Both the PM and gas-phase pollutants exhibited diurnal
cycles indicative of anthropogenic sources, and the ratio of benzene to toluene indicated the influence of fresh,
local emissions. Comparison of canister samples taken over the course of a day showed similarities which would
seem to imply an urban background level for many NMVOC species. In addition to the secondary inorganic
aerosol, a significant fraction of OC was attributed to secondary organic aerosol (18-63%) in the CMB analysis.
The influence of vegetation and biogenic emissions was demonstrated in the canister sample analysis, as
well as the CMB results where vegetative detritus comprised one of the larger sources contributing to the OC
fraction ranging from 11 to 20%. While the detected mixing ratios of the biogenic NMVOCs did not contribute
significantly to the total NMVOC mixing ratio, the role in e.g., ozone formation, assessed by calculating OH
reactivity, was much more significant. Biogenics and their oxidation products accounted for 31% of the OH
reactivity at the urban background station in Neukölln and 75% at the urban park location (Plänterwald),
demonstrating the importance of urban parks for biogenic emissions. These contributions from biogenics were
higher than those found at comparable urban background locations in London and Paris. This is likely linked to
the relatively high amount of land surface area in Berlin which is covered by vegetated areas (34%). It should
however, be acknowledged that only a subset of the total NMVOCs were measured. If all 'missing' NMVOCs
were measured it could influence our results, including the contribution of biogenics and other compound classes
to the calculated OH reactivity.

As an outlook, future research could build on this work to include additional analysis of PTR-MS data

using positive matrix factorization to investigate the sources influencing NMVOC concentrations at the Neukölln
location, as well as modeling studies to gain greater insight as to the impact of urban vegetation on ozone
formation, both yielding further insight into the importance of biogenic VOCs in urban environments.

## 5    Data availability

The datasets generated during and/or analysed during the current study are available from the corresponding
author on request.

## 6    Acknowledgements

This work was hosted by IASS Potsdam, with financial support provided by the Federal Ministry of Education
and Research of Germany (BMBF) and the Ministry for Science, Research and Culture of the State of
Brandenburg (MWFK). The authors would like to thank Hugo Denier van der Gon and Jeroen Kuenen (TNO)
for providing information pertaining to the TNO-MACCIII inventory and Friderike Kuik for the Berlin
emissions processing; Christoph Münkel from Vaisala GmbH, Hamburg for support with ceilometer CL51 data
analyses to determine mixing layer heights; Wolfram Birmili (UBA), Alfred Wiedensohler, and Kay Weinhold
(TROPOS) for discussions informing the particle measurements, colleagues at the IASS for their support of the
campaign and discussions that helped shape the manuscript. The authors gratefully acknowledge the NOAA Air
Resources Laboratory (ARL) for the provision of the HYSPLIT transport and dispersion model and/or READY
website (http://www.ready.noaa.gov) used in this publication. Boris Bonn highly acknowledges a grant from the
IASS to support the studies.

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

**Figure Captions:**

**Figure 1.** Location of the measurement station (MC042) and measurement van in Neukölln, Berlin. Maps show increasingly larger scale. The 'x's indicate sampling locations, with MC220 and MC143 indicating stations that record traffic counts. Map images from OpenStreetMap.

**Figure 2.** Time series of air pollutant concentrations, meteorological data, and benzene/toluene ratio measured as part of BLUME at the Neukölln station during the BAERLIN2014 campaign.

**Figure 3.** Time series of particulate matter mass, particle number, and lung depositable surface area concentrations measured at the Neukölln station during the BAERLIN2014 campaign. (a) BLUME PM10, (b) Grimm 1.108 PM10, (c) Grimm 1.108 PM2.5, (d) Grimm 1.108 PM1, (e) Grimm 1.108 PN, (f) Grimm 5.416 PN, (g) Grimm 5.403 PN, (h) NSAM LDSA. Units given in the y-axis label.

**Figure 4.** Mean diurnal cycles of the (top) particle number and (bottom) particle volume distributions at Neukölln. Legends show particle size bin range in nm.

**Figure 5.** Mean diurnal cycle of the particle number concentration by diameter.

**Figure 6.** Mean fractional contribution to mixing ratio (left column) and OH reactivity (right column) by compound class, based on a total mixing ratio or OH reactivity calculated from 57 compounds for 5 sampling locations throughout the city. Total number of canister samples for each location are Neukölln (18), Altlandsberg (10), Plänterwald (11), Tiergarten Tunnel (9), and the AVUS (2). The individual compounds included in each class are available in the SI.

**Figure 7.** Comparison between VOC measurements in this study and comparable previous work from June-August of 1996 (Thijsse et al., 1999). Compound classes are distinguished by color. Sampling locations by character.

**Figure 8.** Bulk composition analysis results from the PM10 filter samples, presented by filter groups, where GRA=Group A, GRB=Group B, etc. and B17, B19, B30 are individual filters. More information on the filter groups, including a some basic composition information and backtrajectory origin can be found in Table 3.

**Figure 9.** Molecular marker analysis results from the PM10 filter samples, presented by filter groups, where GRA=Group A, GRB=Group B, etc. and B17, B19, B30 are individual filters. More information on the filter groups, including a some basic composition information and backtrajectory origin can be found in Table 3.

**Figure 10.** Source contributions attributed to the OC fraction of the PM10 filter samples by filter groups, where GRA=Group A, GRB=Group B, etc. and B17, B19, B30 are individual filters. More information on the filter groups, including a some basic composition information and backtrajectory origin can be found in Table 3.

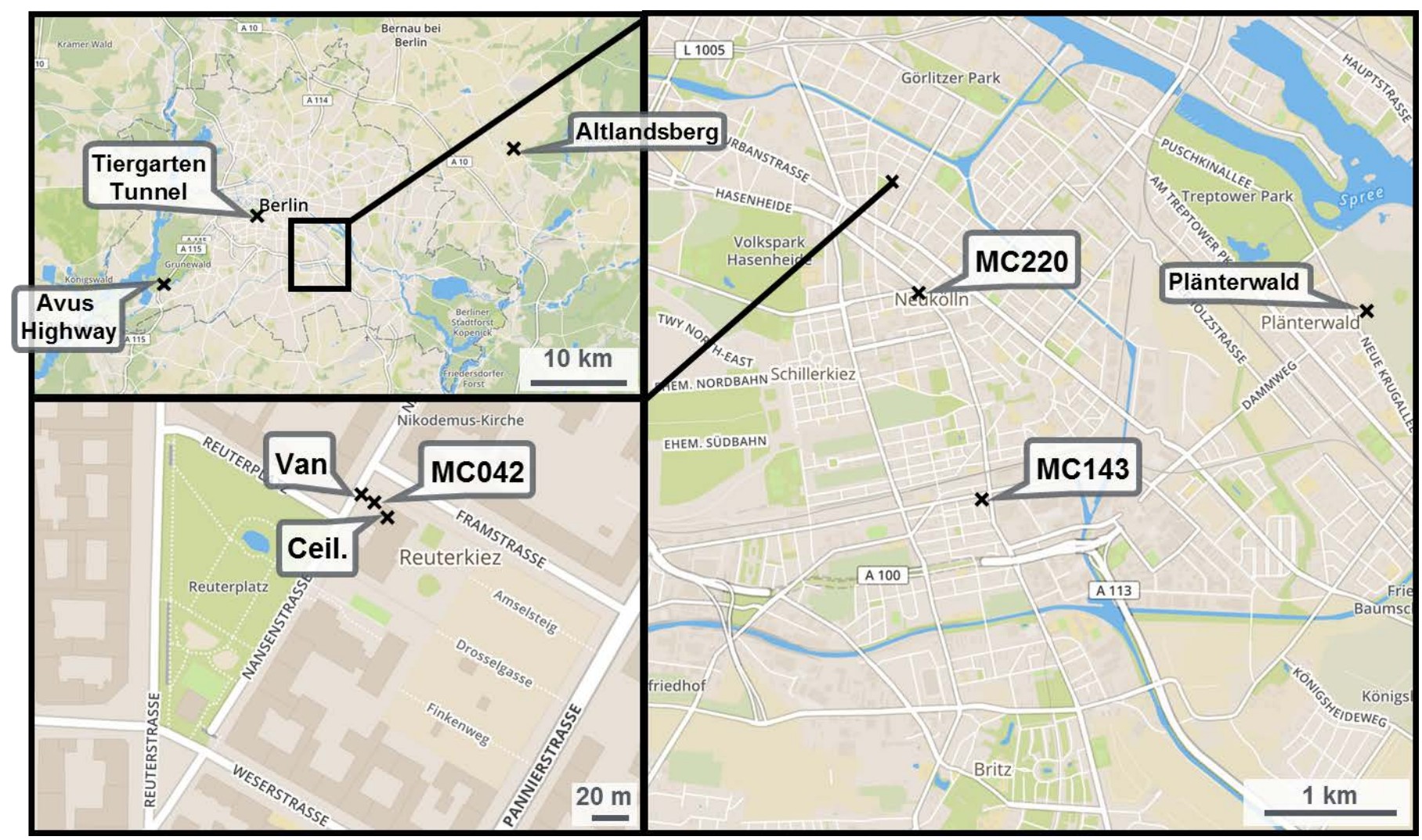

**Figure 1.** Location of the measurement station (MC042) and measurement van in Neukölln, Berlin. Maps show increasingly larger scale. The 'x's indicate sampling locations, with MC220 and MC143 indicating stations that record traffic counts. Map images from OpenStreetMap.

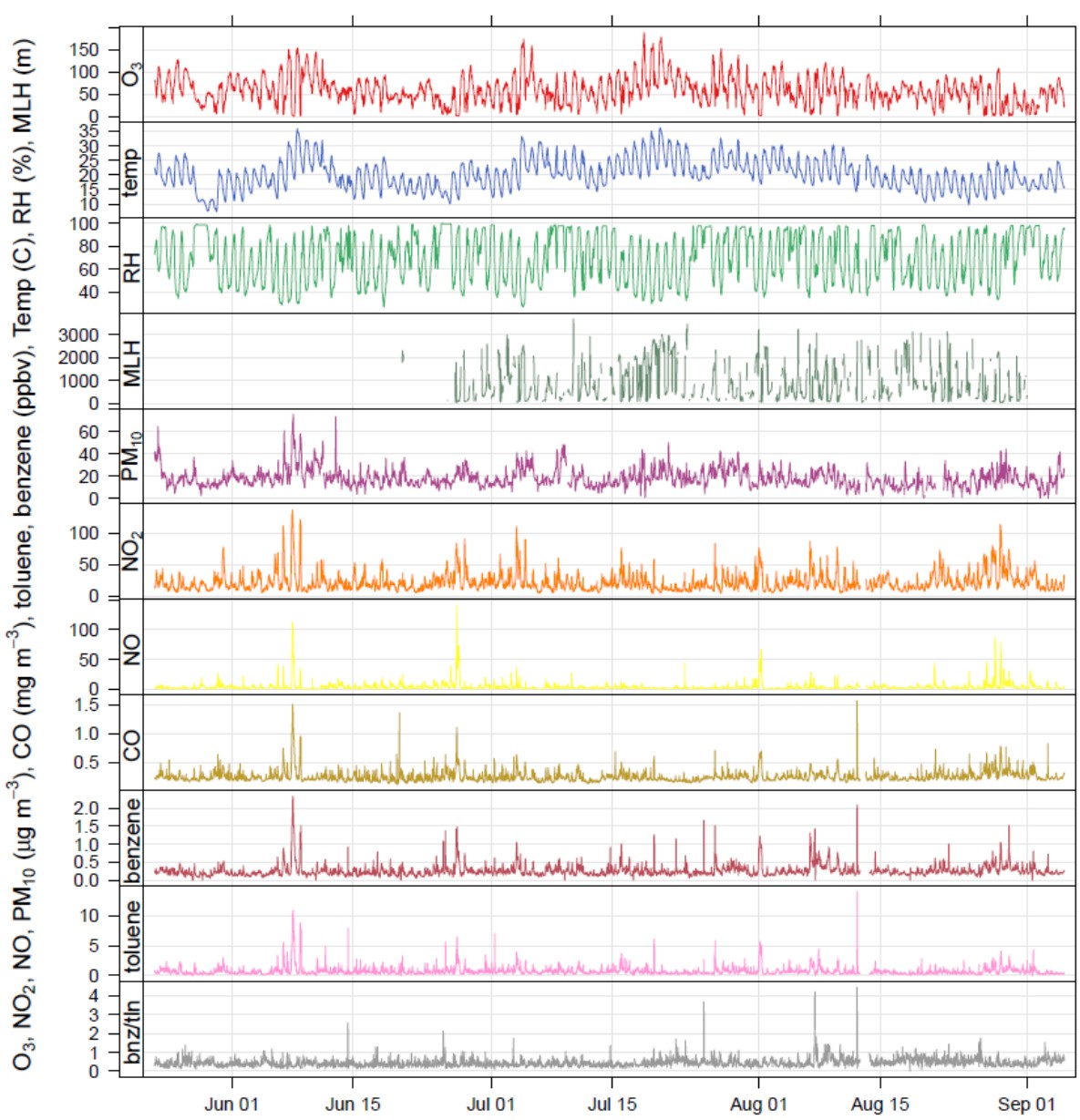

**Figure 2.** Time series of air pollutant concentrations, meteorological data, and benzene/toluene ratio measured as part of BLUME at the Neukölln station during the BAERLIN2014 campaign.

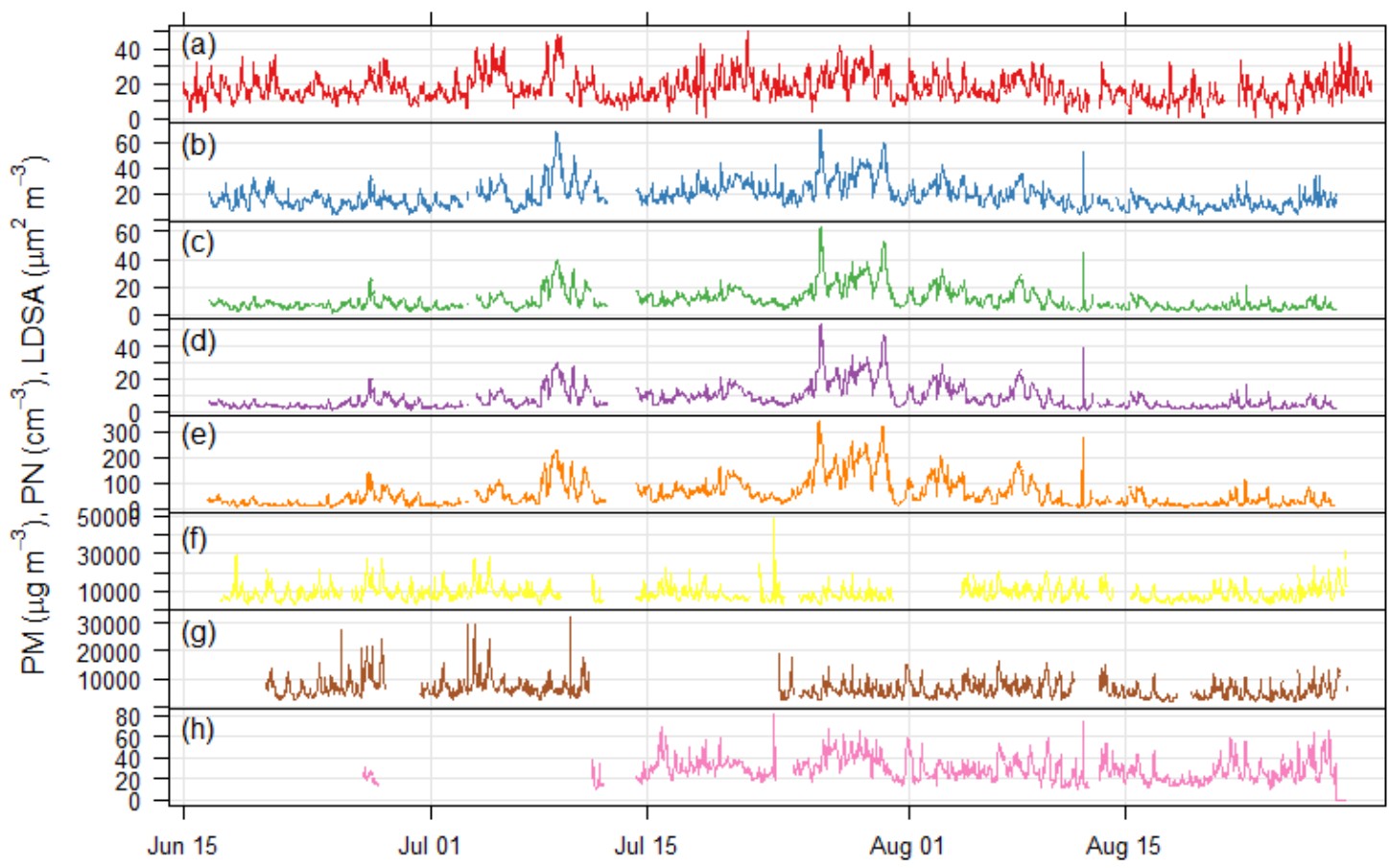

**Figure 3.** Time series of particulate matter mass, particle number, and lung depositable surface area concentrations measured at the Neukölln station during the BAERLIN2014 campaign. (a) BLUME PM10, (b) Grimm 1.108 PM10, (c) Grimm 1.108 PM2.5, (d) Grimm 1.108 PM1, (e) Grimm 1.108 PN, (f) Grimm 5.416 PN, (g) Grimm 5.403 PN, (h) NSAM LDSA. Units given in the y-axis label.

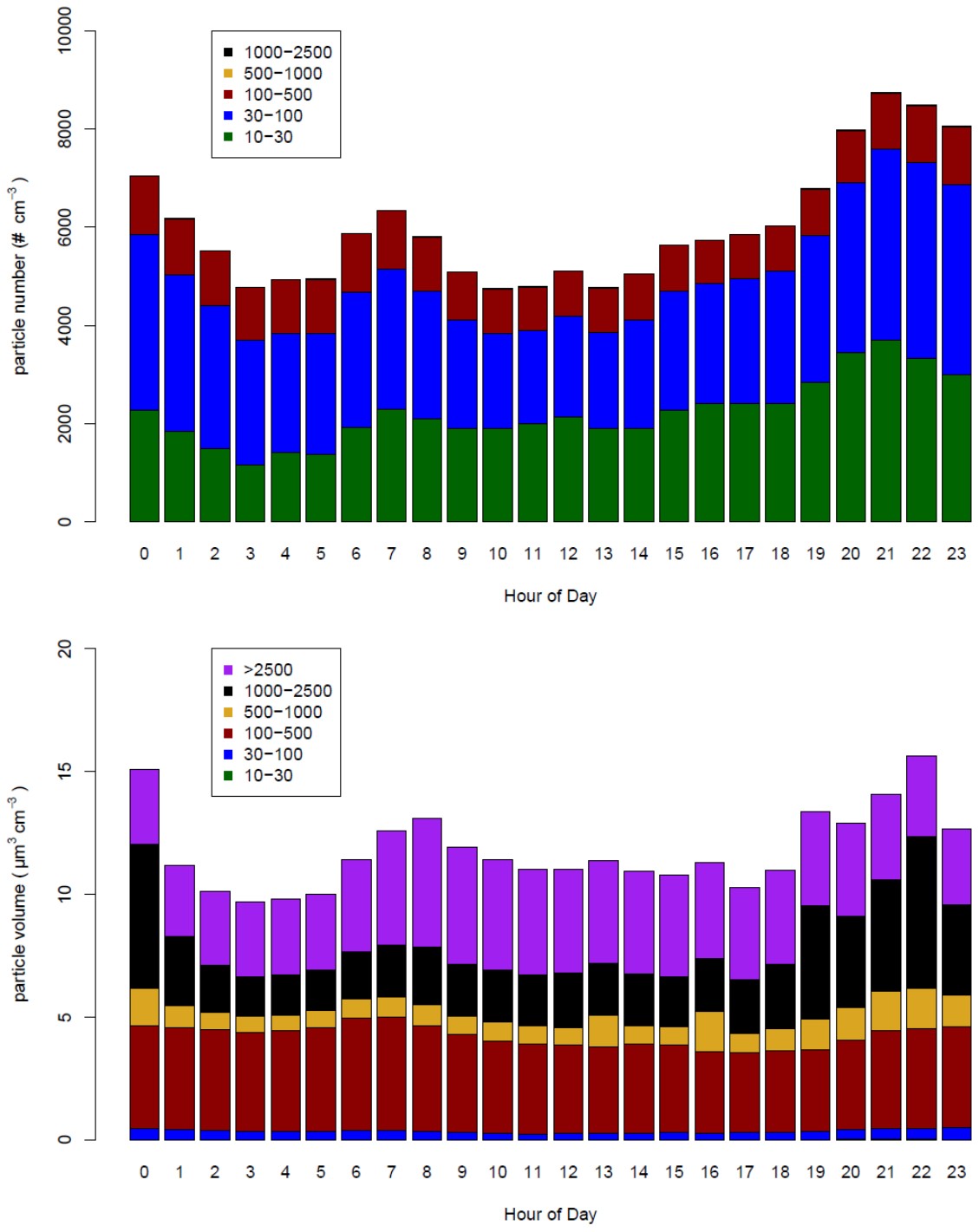

**Figure 4.** Mean diurnal cycles of the (top) particle number and (bottom) particle volume distributions at Neukölln. Legends show particle size bin range in nm.

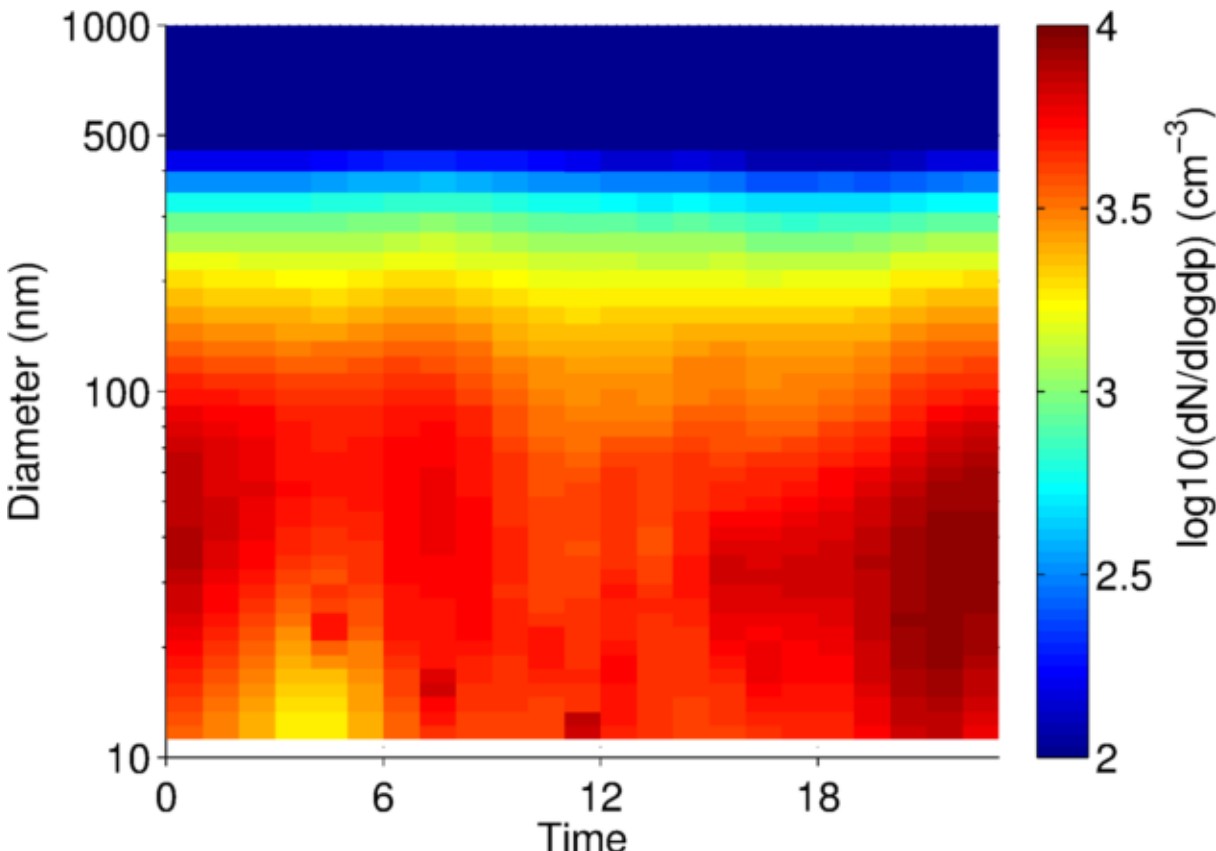

**Figure 5.** Mean diurnal cycle of the particle number concentration by diameter.

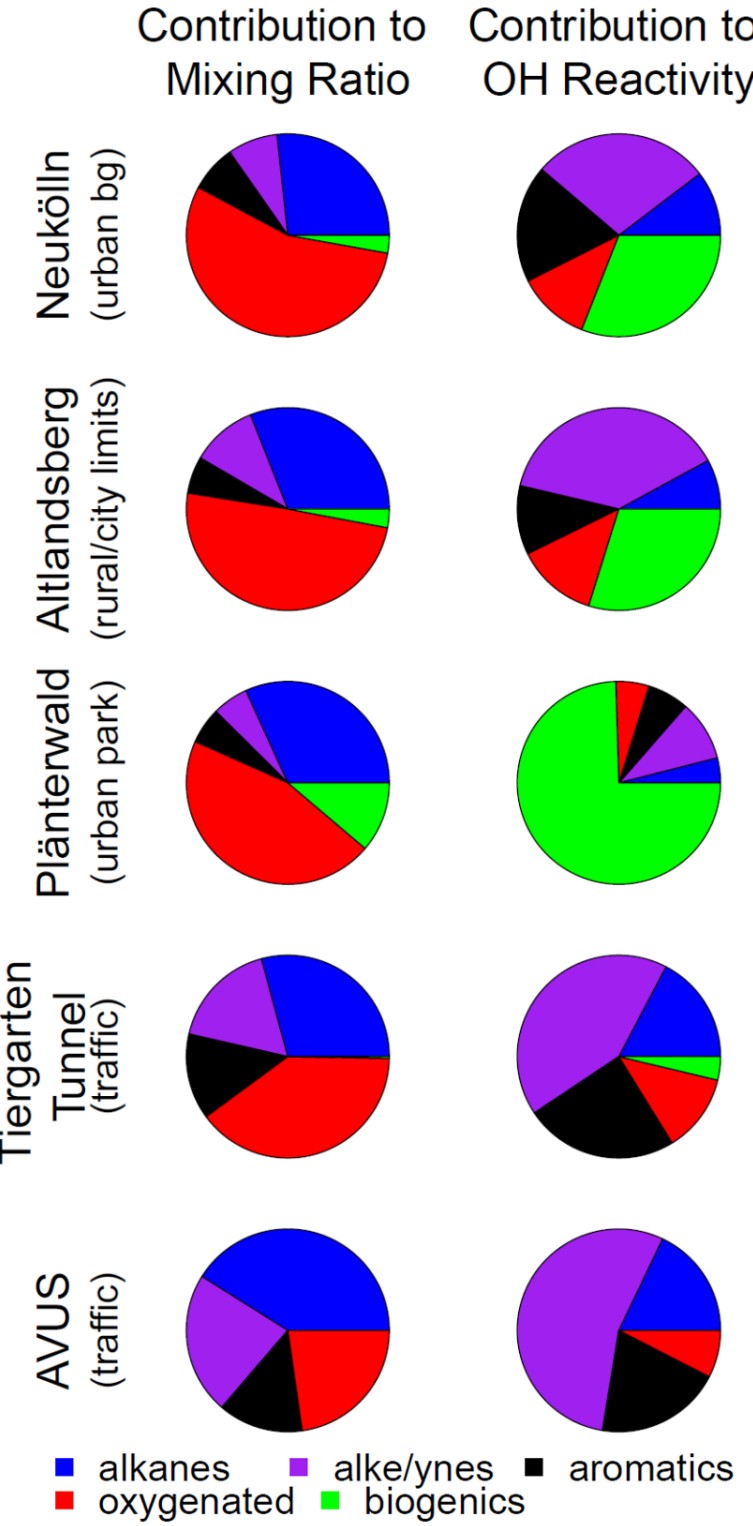

**Figure 6.** Mean fractional contribution to mixing ratio (left column) and OH reactivity (right column) by compound class, based on a total mixing ratio or OH reactivity calculated from 57 compounds for 5 sampling locations throughout the city. Total number of canister samples for each location are Neukölln (18), Altlandsberg (10), Plänterwald (11), Tiergarten Tunnel (9), and the AVUS (2). The individual compounds included in each class are available in the SI. For more information on the site classification, see Table 2.

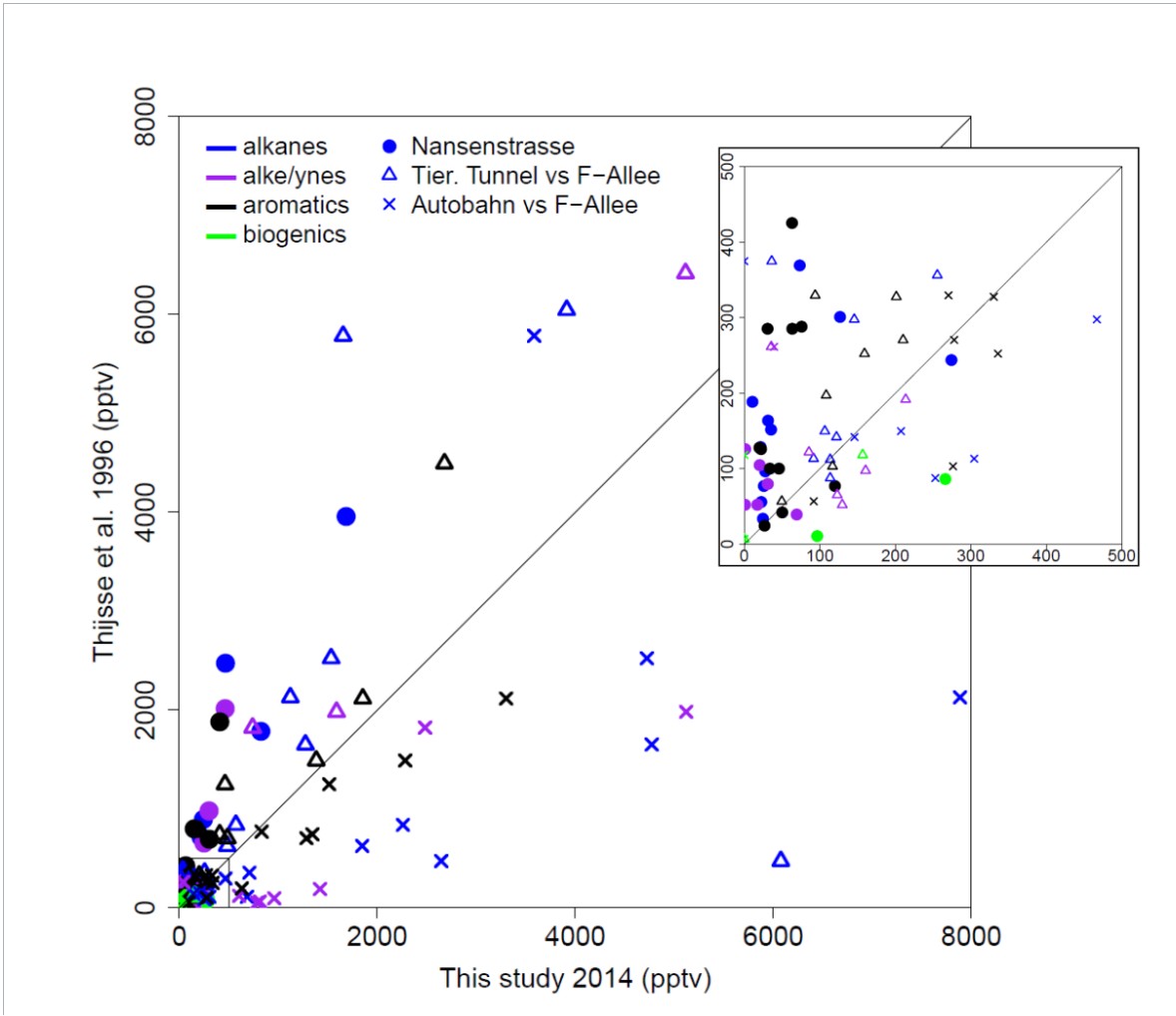

**Figure 7.** Comparison between VOC measurements in this study and comparable previous work from June-August of 1996 (Thijsse et al., 1999). Compound classes are distinguished by color. Sampling locations by character.

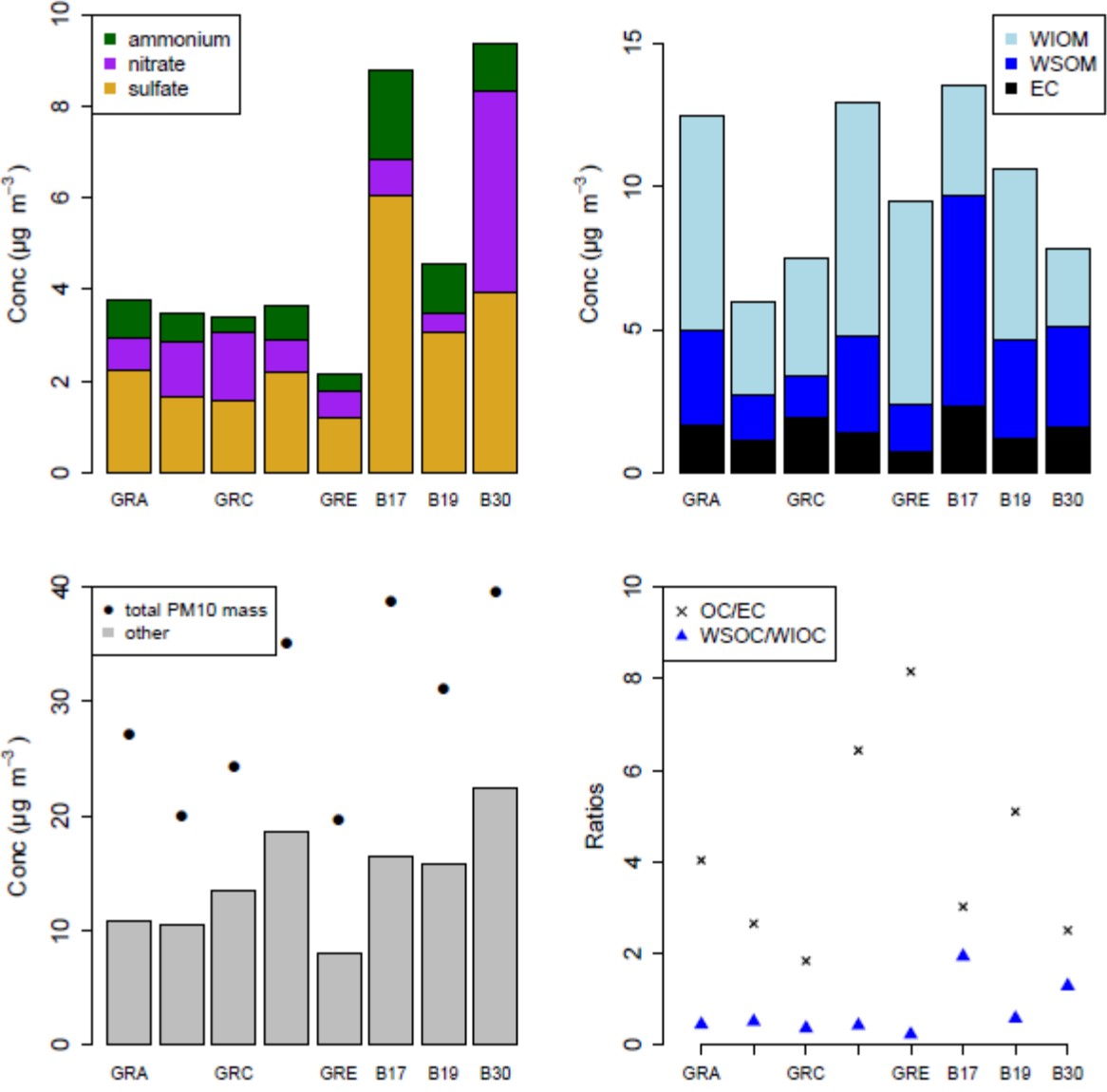

**Figure 8.** Bulk composition analysis results from the PM10 filter samples, presented by filter groups, where GRA=Group A, GRB=Group B, etc. and B17, B19, B30 are individual filters. More information on the filter groups, including a some basic composition information and backtrajectory origin can be found in Table 3.

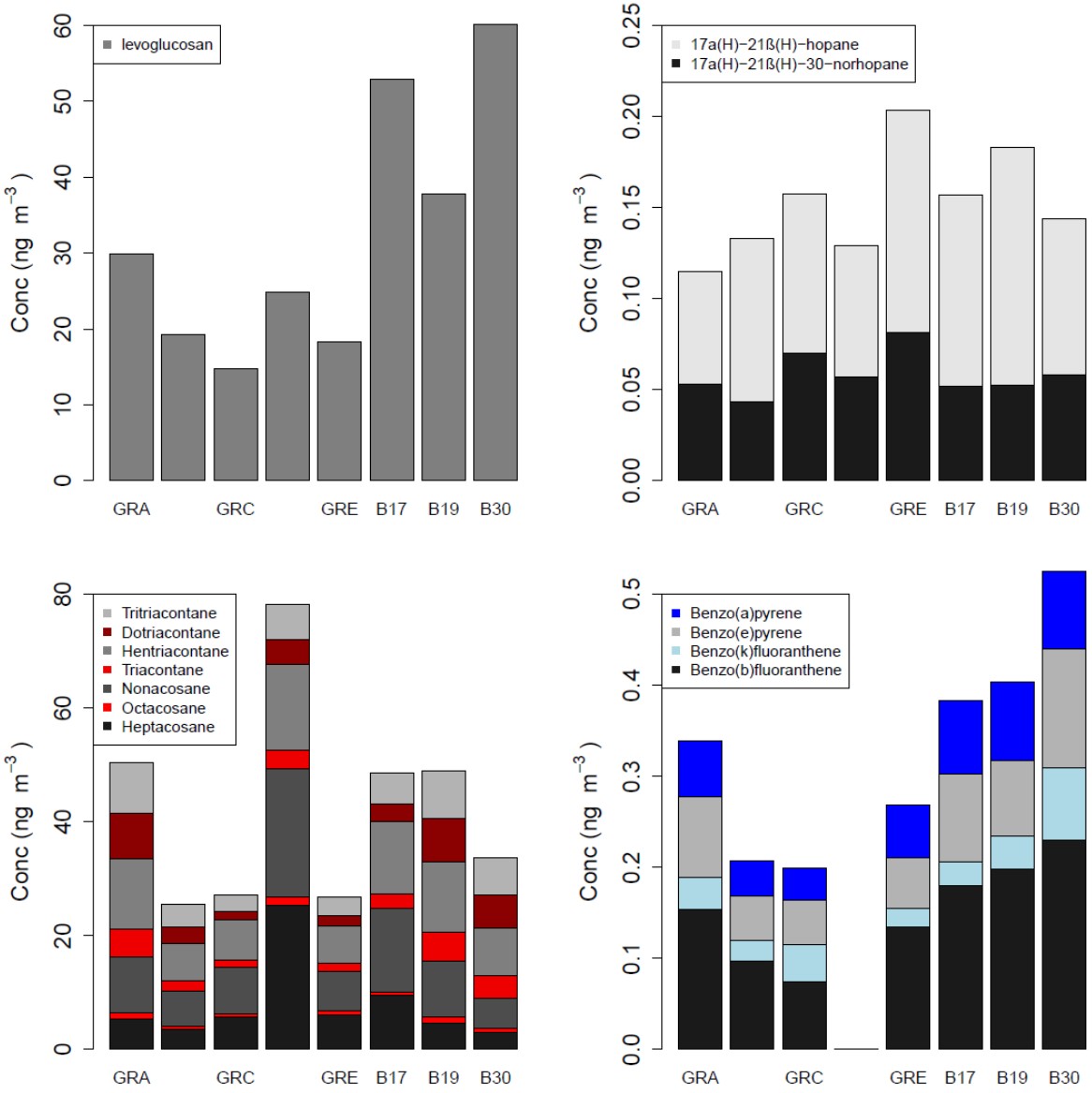

**Figure 9.** Molecular marker analysis results from the PM10 filter samples, presented by filter groups, where GRA=Group A, GRB=Group B, etc. and B17, B19, B30 are individual filters. More information on the filter groups, including a some basic composition information and backtrajectory origin can be found in Table 3.

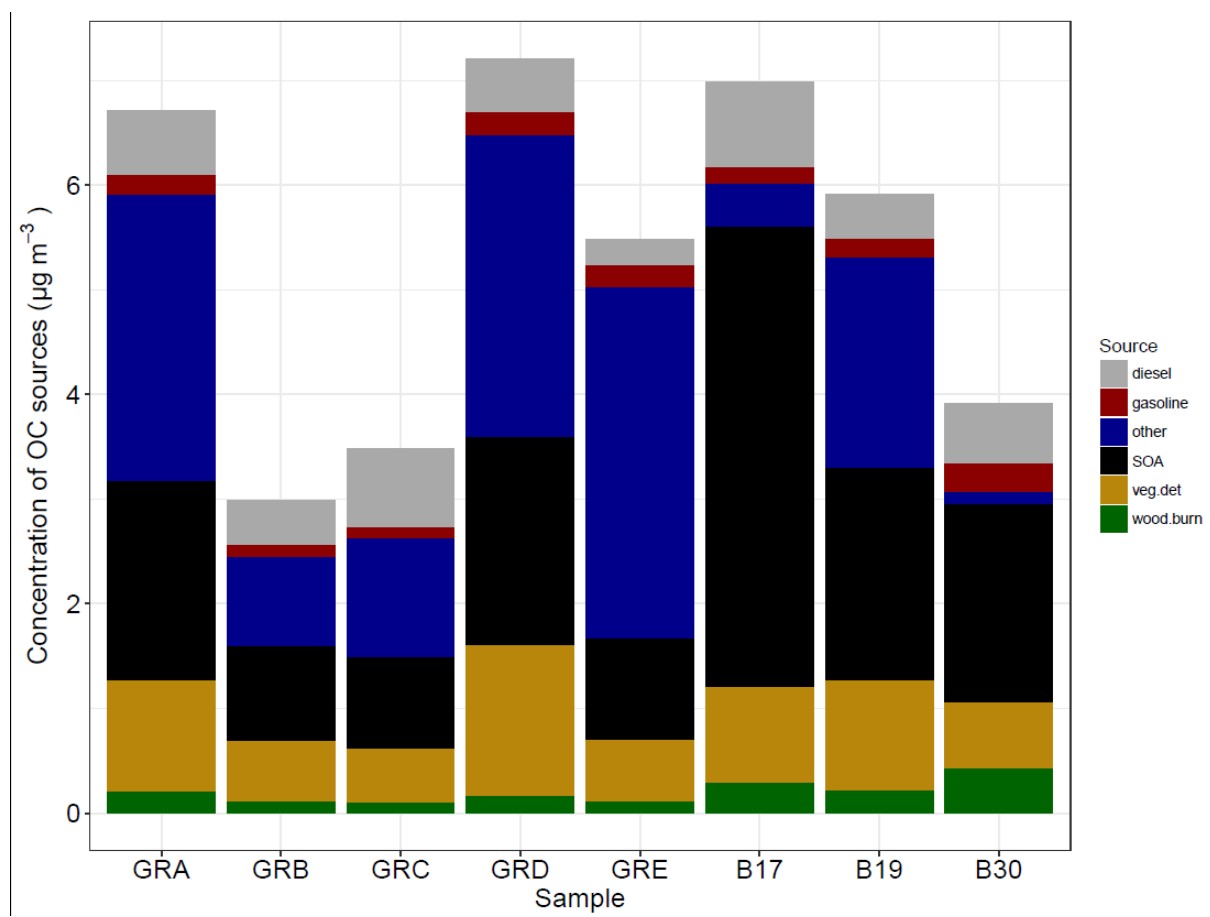

**Figure 10.** Source contributions attributed to the OC fraction of the PM10 filter samples by filter groups, where GRA=Group A, GRB=Group B, etc. and B17, B19, B30 are individual filters. More information on the filter groups, including a some basic composition information and backtrajectory origin can be found in Table 3.

**Table 1.** List of participating institutions and instruments deployed at the urban background site in Berlin (Nansenstrasse).

| Institution | Instrument | Parameters | References |
|---|---|---|---|
| Berlin Senate | Leckel GmbH SEQ47/50 (x1) | $PM_{10}$ | DIN EN 16450:2015-10; Beuth, 2015 |
| | Horiba APNA-370 Air Pollution Monitor | $NO_x$, NO (measured directly); $NO_2$ (inferred) | DIN EN 14211:2005; Verbraucherschutz, 2010 |
| | Horiba APOA-370 Air Pollution Monitor | $O_3$ | DIN EN 14625:2005; Verbraucherschutz, 2010 |
| | Horiba APMA-370 Air Pollution Monitor | CO | DIN EN 14626:2005; Verbraucherschutz, 2010 |
| | AMA Instruments GC5000 BTX | Benzene, toluene | DIN EN 14662:2005; Verbraucherschutz, 2010 |
| KIT | Vaisala CL51 Ceilometer | Mixing layer height | Emeis et al., 2007; Münkel et al., 2007; Wiegner et al., 2014 |
| UBA | GRIMM 1.108 | Particle number and size distribution (350-22500 nm), 15 size bins | Görner et al. 2012 |
| | GRIMM 5.403 | Particle number and size distribution (10-1100 nm), 44 size bins | Heim et al., 2004 |
| | GRIMM 5.416 | Total particle number (4-3000 nm) | Helsper et al., 2008; Wiedensohler et al., 2017 |
| | NSAM | Particle surface area (10-1000 nm) | Kaminski et al., 2013; VDI 2017 |
| IASS | PTR-MS | NMVOCs (for a complete list of m/z see Table S1) | Bourtsoukidis et al. 2014 |
| FZJ | Canister samples | NMVOCs (for a list of compounds, see Table 8 in Bonn et al., 2016, or for the 57 compounds included in this analysis the SI) | Urban 2010; Ehlers et al. 2016 |
| | Filter sampling/analysis | $PM_{10}$, mass, EC, OC | Kofahl 2012; Ehlers 2013 |
| FMI-Helsinki | Cartridge samples | Biogenic NMVOCs | Mäki et al. 2017 |
| UW-Madison | Filter analysis | WSOC, WIOC, ions, organic tracers | Yang et al., 2003; Wang et al., 2005; Miyazaki et al., 2011; Villalobos et al., 2015 |

**Table 2.** NMVOC canister sampling locations, site type, and average OH reactivity ($s^{-1}$)

| | Location type | alkanes | alkenes | aromatics | oxygenated | biogenics | total |
|---|---|---|---|---|---|---|---|
| Neukölln† | Urban background station | 0.27±0.10 | 0.75±0.40 | 0.49±0.29 | 0.29±0.08 | 0.82±0.44 | 2.6±0.68 |
| Altlandsberg | Rural, agricultural area with a small town, partially forested | 0.17±0.10 | 0.83±0.43 | 0.22±0.11 | 0.28±0.17 | 0.65±0.42 | 2.2±0.69 |
| Plänterwald | approx. 1 km$^2$ urban park abutting the Spree river in eastern Berlin | 0.20±0.06 | 0.47±0.14 | 0.33±0.12 | 0.25±0.04 | 3.7±0.90 | 4.9±1.0 |
| Tiergarten Tunnel* | 2.4 km tunnel, major 4-lane city thoroughfare in central Berlin | 2.0±2.2 | 4.4±1.1 | 2.6±1.3 | 1.3±0.70 | 0.39±0.24 | 11±2.5 |
| AVUS* | Highly traficked motorway in western Berlin (traffic jam) | 6.3±3.2 | 19±7.4 | 6.6±1.6 | 2.8±2.3 | 0.00±0.00 | 34±15 |

* automated sampling while driving; all other samples taken from a stationary location.

† 20 minute sampling duration. All other samples had 10 minute sampling duration.

**Table 3.** Basic bulk composition results, ratios, and air mass origin from HYSPLIT. Units are µg m-3 unless otherwise noted. For OC and ED measurement uncertainty is included.

| | Total PM10 | Air mass origin (HYSPLIT) | Total OC (± unc) | Total EC (± unc) | Total Ions* | OC:EC ratio | WSOC of OC (%) | Ions:OC ratio** |
|---|---|---|---|---|---|---|---|---|
| Group A | 27.1 | Germany | 6.7 ± 0.34 | 1.7 ± 0.084 | 5.1 | 4.0 | 31% | 0.56 |
| Group B | 20.0 | central Germany, northern France | 3.0 ± 0.15 | 1.1 ± 0.057 | 5.3 | 2.7 | 34% | 1.2 |
| Group C | 24.4 | North Sea | 3.5 ± 0.17 | 1.9 ± 0.094 | 5.7 | 1.8 | 27% | 0.98 |
| Group D | 35.1 | Baltic | 7.2 ± 0.36 | 1.4 ± 0.069 | 5.0 | 6.4 | 30% | 0.50 |
| Group E | 19.6 | North Sea, Scandinavia, UK | 5.5 ± 0.27 | 0.71 ± 0.035 | 3.2 | 8.1 | 19% | 0.39 |
| B17 | 38.8 | Poland & east | 7.0 ± 0.35 | 2.3 ± 0.12 | 11 | 3.0 | 66% | 1.3 |
| B19 | 31.0 | Poland & north | 5.9 ± 0.30 | 1.2 ± 0.058 | 6.0 | 5.1 | 37% | 0.77 |
| B30 | 39.5 | Germany (northern France) | 3.9 ± 0.20 | 1.6 ± 0.078 | 15 | 2.5 | 56% | 2.4 |

*Ions includes 7 species and is not limited to sulfate, nitrate, and ammonium.

**Ratio of ions (sulfate, nitrate, ammonium) to OC

**Table 4.** Chemical mass balance source apportionment results. Units are µg m-3 unless otherwise noted. Uncertainty is measurement uncertainty, in the case of SOA propagated uncertainty.

| | Total OC (unc) | % OC mass apportioned | measured WSOC (unc) | SOA* (unc) | veg. det. (std error) | wood burn. (std error) | diesel emissions (std error) | gasoline vehicles (std error) | $R^2$ | $\chi^2$ |
|---|---|---|---|---|---|---|---|---|---|---|
| Group A | 6.71± 0.34 | 30.8 | 2.06±0.10 | 1.91±0.11 | 1.07±0.13 | 0.21±0.04 | 0.61±0.06 | 0.19±0.02 | 0.77 | 12.39 |
| Group B | 2.99± 0.15 | 41.2 | 1.00±0.05 | 0.91±0.05 | 0.57±0.07 | 0.12±0.03 | 0.42±0.04 | 0.12±0.02 | 0.8 | 7.7 |
| Group C | 3.48± 0.17 | 42.4 | 0.94±0.05 | 0.87±0.05 | 0.52±0.06 | 0.10±0.02 | 0.74±0.07 | 0.11±0.02 | 0.85 | 5.38 |
| Group D | 7.21± 0.36 | 32.3 | 2.11±0.11 | 1.99±0.11 | 1.44±0.17 | 0.17±0.04 | 0.50±0.05 | 0.22±0.03 | 0.87 | 6.82 |
| Group E | 5.48± 0.27 | 21.2 | 1.05±0.05 | 0.97±0.06 | 0.59±0.07 | 0.12±0.03 | 0.24±0.03 | 0.21±0.02 | 0.77 | 9.78 |
| B17 | 6.99± 0.35 | 31.1 | 4.61±0.23 | 4.40±0.24 | 0.91±0.10 | 0.30±0.07 | 0.81±0.08 | 0.15±0.03 | 0.8 | 7.89 |
| B19 | 5.91± 0.30 | 31.7 | 2.19±0.11 | 2.03±0.12 | 1.05±0.12 | 0.22±0.05 | 0.42±0.04 | 0.18±0.03 | 0.73 | 9.83 |
| B30 | 3.91± 0.20 | 48.6 | 2.21±0.11 | 1.90±0.13 | 0.63±0.08 | 0.44±0.09 | 0.57±0.06 | 0.28±0.04 | 0.76 | 10.17 |

*The SOA contribution was not part of the CMB results, but rather calculated as: unapportioned WSOC (SOA) = measured WSOC – 0.71*apportioned wood burning.