# Peer review of "BAERLIN2014 – stationary measurements and source apportionment at an urban background station in Berlin, Germany"

_Atmospheric Chemistry and Physics, 2017_

## Referee Comment (RC1) · Anonymous Referee #1 · 3 Jan 2018

This manuscript presents a very elaborate description of a part of the results of a 3-months summer campaign in Berlin 2014. VOC and PM10 data of ground-based stations are analyzed. Data are presented with care and in great detail including supplementary material. Plenty of data were produced, which somewhat justifies that they are presented in 2 companion papers, of which the present one is the second. Data are analyzed using various techniques including statistics, comparison with emission inventories, backward trajectories. The results provide a valuable insight into the chemistry of gas and particulate phases of the Berlin urban agglomeration in summer 2014.

[Figure]

Specifically the contribution of biogenic sources to the reactivity of organic material is analyzed. Data are compared with previous, similar studies in other urban areas of the world such as Paris (France) and the Pearl River Delta (China). The style of the presentation is fluid and smooth. For all these reasons, the manuscript should be accepted for publication in ACP. This reviewer has only one remark and some minor editorial comments that the authors should respect when preparing the final version. Unfortunately, the manuscript does not present any novel insight or idea. It follows common lines and techniques. It is more a technical expertise than a scientifically thrilling contribution.

Remark:

There is some mismatch in arguments in that the chemical composition of PM10 is, on the one hand (section 3.5.1), discussed in terms of medium range backward trajectories of air masses and on the other hand (section 3.5.4), discussed in terms of the more local emission inventory of the Berlin area. The authors should emphasize more explicitly the limitations of both of these analyzes.

Editorial comments:

Figs. 4, 8, 9, 10, y-axes units: superscripts ("3") should be formatted as superscripts

Fig. 5: The grey background color seems somewhat awkward. The color scale has no unit. The figure should not have a headline.

Fig. 7: Why do the axes' scales start at mixing ratios below zero? This seems like a standard R graph, which should be optimized.

line 37 and many other places in the manuscript: Replace "ca." by "approx."

line 133: On first occurrence of AVUS, you may say "the so-called AVUS motorway" or similar

line 259: please subscribe the x in NOx

---

## Referee Comment (RC2) · Anonymous Referee #2 · 1 Apr 2018

The study reports on VOC and PM measurements made in Berlin during the BAER-LIN2014 field campaign. A variety of instruments were used to make measurements of VOCs (Canisters, Cartridges, PTR-MS) and PM (filters, particle number, surface area). These measurements are of great value to the atmospheric chemistry community, and provide insights on air pollution in a major European city. The main finding is that biogenic emissions are significant contributors to ozone and PM in Berlin. This is clearly an important finding. However, I have some critiques that hopefully will strengthen the study's main conclusion. Also, at times the manuscript seems to be written as an

overview paper of the campaign, which distracts from the analysis of field measurements. With revisions strengthening the robustness of VOC results, along with streamlining the manuscript to emphasize key results, I believe this paper has the potential to be published in Atmospheric Chemistry & Physics.

General Comments

(1) For estimating OH reactivity, the results appear to be based only on VOC canister samples. Canisters could be missing key VOCs, especially oxygenated compounds, which may contribute significantly to OH reactivity. Karl et al. (2018) recently found a large flux of oxygenated VOC emissions in a European city, around half of the total VOC flux, including more highly oxidized OVOCs. The more oxidized OVOCs, which Karl et al. measured using a PTR-ToF-MS, are not on the list of compounds listed in Table S1. Therefore, this study likely presents an upper bound estimate of biogenic VOC emissions on OH reactivity, due to missing OVOCs. Discussing potential gaps in canister sampling systems in measuring VOCs and how it could affect results of this study is warranted.

Karl, T., et al. (2018). "Urban flux measurements reveal a large pool of oxygenated volatile organic compound emissions." Proceedings of the National Academy of Sciences of the United States of America 115(6): 1186-1191.

(2) Since the analysis appears to focus on canister samples for measurements of VOCs and in estimating of OH reactivity. It is not clear why measurements by cartridge samples (Section 2.2.2.) and PTR-MS (Section 2.2.3.) are included in the manuscript, other than to show that such measurements were made in BAERLIN2014. If these measurement systems are to be included, a more thorough evaluation of their VOC data is needed. By contrast, I found the PM instruments described to be well discussed and presented in the Results & Discussion section.

(3) Lines 452-467. The lack of agreement between the PTR-MS and VOC canister sampling analysis is disconcerting. While it is true that the PTR-MS may lack specificity of individual compounds at a given m/z, some masses have been fairly well characterized now in urban air, including OVOCs (e.g., acetaldehyde, acetone, MEK, and methanol), aromatics (e.g., benzene and toluene), and monoterpenes (see Warneke et al., 2007). The way this paragraph is written, it appears to dismiss the PTR-MS measurements. However, there are also questions about sampling artifacts of key classes of compounds by canisters. For example, Lerner et al. (2017) report significant sampling artifacts present in canister samples of OVOCs and heavy aromatics (C9+). The analysis of VOC measurements could be strengthened by a more thorough evaluation for why differences are observed in the PTR-MS and canister samples, and by leveraging measurements from the two systems later in the analysis, rather than only highlighting results from the canister samples. The discussion mainly focuses on correlations, but are there any systematic biases in concentrations between the two instruments?

Warneke, C., et al. (2007). "Determination of urban volatile organic compound emission ratios and comparison with an emisions database." Journal of Geophysical Research-Atmospheres 112: D10S47.

Lerner, B. M., et al. (2017). "An improved, automated whole air sampler and gas chromatography mass spectrometry analysis system for volatile organic compounds in the atmosphere." Atmospheric Measurement Techniques 10(1): 291-313.

Specific Comments

(4) Line 165. It is not clear here how terpenes are affected by canister transport and storage.

(5) Line 435. It is not clear what the "BLUME network" is. Some description about what this measurement is would be helpful.

(6) Line 440. It is not clear which instrument is located at street-level. In the following discussion, it is implied that the PTR-MS is at street-level, but not explicitly.

(7) Line 464. What is m/z 9? This molecule would be smaller than carbon, so not a VOC.

(8) Line 499. There are several points below the 1:1, suggesting increases in mixing ratios. It would be interesting to highlight what these compounds are, and whether their lack of decrease/increase in concentration is consistent with the literature.

(9) Line 522. Why is limonene not included under the biogenic category when it is measured (Table S1)? Not including limonene might understate the biogenic contribution. It would also help to break down the OH reactivity between isoprene, a-pinene, and b-pinene for the Neukolln and Altlandsberg sites. Some terpenes may be manmade and not biogenic (Derwent et al., 2007).

Derwent, R. G., et al. (2007). "Photochemical ozone creation potentials (POCPs) for different emission sources of organic compounds under European conditions estimated with a Master Chemical Mechanism." Atmospheric Environment 41(12): 2570-2579.

(10) Section 3.5.1. While I do not disagree with any of the statements made here, it was not clear by the end of the section what the new insights were. Also, this section could benefit from describing the bulk composition first across all samples, and provide better context for the back-trajectory analysis.

(11) Line 696 – 703. Are the diesel and gasoline vehicle contributions from POA only? If so, a caveat may be warranted here that secondary PM from gasoline and diesel vehicles are not included, which are potentially important sources of PM from transportation (see Gordon et al., 2014).

Gordon, T. D., et al. (2014). "Secondary organic aerosol formation exceeds primary particulate matter emissions for light-duty gasoline vehicles." Atmospheric Chemistry and Physics 14(9): 4661-4678.

Gordon, T. D., et al. (2014). "Secondary organic aerosol production from diesel vehicle exhaust: Impact of aftertreatment, fuel chemistry and driving cycle." Atmospheric Chemistry and Physics 14(9): 4643-4659.

(12) Line 709. It is not clear how high concentrations of inorganics support the finding of high amounts of SOA. Please describe in further detail.

(13) Line 750. While I do not dispute that biogenic VOCs are reactive and have an outsized contribution on OH reactivity, I believe caveats are needed here that missing VOCs not measured could affect the BVOC contributions presented here.

(14) Figure 6. It would be helpful to label which sites are traffic-dominated, urban background, and urban park under the name of each site.

---

## Author Comment (AC1) · 1 May 2018

Interactive comments on "BAERLIN2014 – stationary measurements and source apportionment at an urban background station in Berlin, Germany" by Erika von Schneidemesser et al.

(All changes also shown in the attached track changes manuscript.)

Anonymous Referee #1

[Figure]

This manuscript presents a very elaborate description of a part of the results of a 3-months summer campaign in Berlin 2014. VOC and PM10 data of ground-based stations are analyzed. Data are presented with care and in great detail including supplementary material. Plenty of data were produced, which somewhat justifies that they are presented in 2 companion papers, of which the present one is the second. Data are analyzed using various techniques including statistics, comparison with emission inventories, backward trajectories. The results provide a valuable insight into the chemistry of gas and particulate phases of the Berlin urban agglomeration in summer 2014.

Specifically the contribution of biogenic sources to the reactivity of organic material is analyzed. Data are compared with previous, similar studies in other urban areas of the world such as Paris (France) and the Pearl River Delta (China). The style of the presentation is fluid and smooth. For all these reasons, the manuscript should be accepted for publication in ACP. This reviewer has only one remark and some minor editorial comments that the authors should respect when preparing the final version. Unfortunately, the manuscript does not present any novel insight or idea. It follows common lines and techniques. It is more a technical expertise than a scientifically thrilling contribution.

Response: We thank the referee for the generally very positive assessment of the paper. We do think that the results are novel in the sense that they considerably advance our understanding of the atmospheric chemistry interactions, pollution levels, and sources in an important European city. However, we agree that there were no major surprises found in the observations yet that would warrant a highlight paper, which is why we are pleased to publish this as a disciplinary paper in ACP.

Remark: There is some mismatch in arguments in that the chemical composition of PM10 is, on the one hand (section 3.5.1), discussed in terms of medium range backward trajectories of air masses and on the other hand (section 3.5.4), discussed in terms of the more local emission inventory of the Berlin area. The authors should emphasize more explicitly the limitations of both of these analyzes.

Response: The reviewer brings up a good point. Some text has been added at the end of section 3.5.3 to highlight these differences and limitations. Furthermore, the limitations of the comparison to the EI is also discussed in more detail in section 3.5.4. The added text reads: "It should be noted that ambient air samples include contributions from both local sources as well as emissions that have been transported from locations further away. While the back trajectory analysis is more relevant for interpreting the influence of emissions from the surrounding region, a comparison to the Berlin emission inventory reflects on the influence of local source contributions. Both play a role, but neither capture the complete picture, with limitations in both cases, as discussed further below."

Editorial comments: Figs. 4, 8, 9, 10, y-axes units: superscripts ("3") should be formatted as superscripts

Response: Done.

Fig. 5: The grey background color seems somewhat awkward. The color scale has no unit. The figure should not have a headline.

Response: These aspects of the figure have been changed as suggested.

Fig. 7: Why do the axes' scales start at mixing ratios below zero? This seems like a standard R graph, which should be optimized.

Response: The reviewer is correct, this was an oversight and has been corrected.

line 37 and many other places in the manuscript: Replace "ca." by "approx."

Response: Done.

line 133: On first occurrence of AVUS, you may say "the so-called AVUS motorway" or similar

Response: Done.

line 259: please subscribe the x in NOx

Response: While no occurrence of NOx was found on line 259, the manuscript was searched for occurrences of NOx and anywhere where the x was not subscripted was corrected.

Anonymous Referee #2

The study reports on VOC and PM measurements made in Berlin during the BAER-LIN2014 field campaign. A variety of instruments were used to make measurements of VOCs (Canisters, Cartridges, PTR-MS) and PM (filters, particle number, surface area). These measurements are of great value to the atmospheric chemistry community, and provide insights on air pollution in a major European city. The main finding is that biogenic emissions are significant contributors to ozone and PM in Berlin. This is clearly an important finding.

However, I have some critiques that hopefully will strengthen the study's main conclusion. Also, at times the manuscript seems to be written as an overview paper of the campaign, which distracts from the analysis of field measurements. With revisions strengthening the robustness of VOC results, along with streamlining the manuscript to emphasize key results, I believe this paper has the potential to be published in Atmospheric Chemistry & Physics.

Response: We thank the reviewer for the comments and believe that the suggested revisions have strengthened the paper. The paper was indeed written to be a form of overview paper, but with an attempt to go beyond a simple overview of the campaign to provide some important findings from data analyses as well. We hope that by addressing the reviewer comments we have found a sufficient balance.

General Comments (1) For estimating OH reactivity, the results appear to be based only on VOC canister samples. Canisters could be missing key VOCs, especially oxygenated compounds, which may contribute significantly to OH reactivity. Karl et al.

(2018) recently found a large flux of oxygenated VOC emissions in a European city, around half of the total VOC flux, including more highly oxidized OVOCs. The more oxidized OVOCs, which Karl et al. measured using a PTR-ToF-MS, are not on the list of compounds listed in Table S1. Therefore, this study likely presents an upper bound estimate of biogenic VOC emissions on OH reactivity, due to missing OVOCs. Discussing potential gaps in canister sampling systems in measuring VOCs and how it could affect results of this study is warranted.

Karl, T., et al. (2018). "Urban flux measurements reveal a large pool of oxygenated volatile organic compound emissions." Proceedings of the National Academy of Sciences of the United States of America 115(6): 1186-1191.

Response: The reviewer makes an important point. We have added text to section 3.4 in two places to address this limitation, as well as integrating the Karl et al. study. The added text reads: "The contribution from oxygenated compounds, despite their substantial contribution to mixing ratio, ranged from only 5-13% of OH reactivity. That said, only 6 oxygenated NMVOCs (of 57 total NMVOCs) were included here, and a recent study by Karl et al., (2018) found an appreciably greater fraction of oxygenated NMVOCs in urban areas than previous studies identified. The molar flux of oxygenated NMVOCs being actively emitted into the urban atmosphere from measurements in Europe was found to be $56 \pm 10\%$ relative to the total NMVOC flux (Karl et al., 2018), which indicates that a much larger contribution from oxygenated NMVOCs is possible if different measurement techniques are used." and "Finally, while the 57 NMVOCs included here to calculate OH reactivity were chosen to facilitate comparison to previous studies, a more exhaustive list could change the picture. For example, as mentioned above, the limited number of oxygenated NMVOCs measured would likely lessen the contributions of the other compound classes."

(2) Since the analysis appears to focus on canister samples for measurements of VOCs and in estimating of OH reactivity. It is not clear why measurements by cartridge samples (Section 2.2.2.) and PTR-MS (Section 2.2.3.) are included in the manuscript,

other than to show that such measurements were made in BAERLIN2014. If these measurement systems are to be included, a more thorough evaluation of their VOC data is needed. By contrast, I found the PM instruments described to be well discussed and presented in the Results & Discussion section.

Response: We thank the reviewer for the comment on the PM information presentation and discussion. As to the VOC measurements, this was in part written as an overview paper and further analysis, specifically PMF analysis that has already been conducted, is planned to be published based on the PTR-MS data. However, including that here would make the already long manuscript, way too long. For this reason, even though the PTR-MS data is not discussed in significant detail, we wanted to include it here. It could be mentioned that such analysis is planned, but we are generally not in favor of doing so unless such work is already at or very near the submitted stage. The point about the cartridge samples is well taken and given later comments about the relative importance of bVOCs, additional discussion of the data from the cartridge samples, which were taken to specifically address a wider range of bVOCs, was added in section 3.4. For changes please see the track changes version of the manuscript, as well as the answer to comment 9.

(3) Lines 452-467. The lack of agreement between the PTR-MS and VOC canister sampling analysis is disconcerting. While it is true that the PTR-MS may lack specificity of individual compounds at a given m/z, some masses have been fairly well characterized now in urban air, including OVOCs (e.g., acetaldehyde, acetone, MEK, and methanol), aromatics (e.g., benzene and toluene), and monoterpenes (see Warneke et al., 2007). The way this paragraph is written, it appears to dismiss the PTR-MS measurements. However, there are also questions about sampling artifacts of key classes of compounds by canisters. For example, Lerner et al. (2017) report significant sampling artifacts present in canister samples of OVOCs and heavy aromatics (C9+). The analysis of VOC measurements could be strengthened by a more thorough evaluation for why differences are observed in the PTR-MS and canister samples, and by leveraging measurements from the two systems later in the analysis, rather than only highlighting results from the canister samples. The discussion mainly focuses on correlations, but are there any systematic biases in concentrations between the two instruments?

Warneke, C., et al. (2007). "Determination of urban volatile organic compound emission ratios and comparison with an emisions database." Journal of Geophysical Research-Atmospheres 112: D10S47.

Lerner, B. M., et al. (2017). "An improved, automated whole air sampler and gas chromatography mass spectrometry analysis system for volatile organic compounds in the atmosphere." Atmospheric Measurement Techniques 10(1): 291-313.

Response: The reviewer brings up a good point. To a certain extent this section may have been written without enough specificity, since most previous work when comparing different instruments are done as part of inter-comparisons where e.g. the same inlet system is used to ensure comparable air masses, etc. That is not the case here and therefore shouldn't be compared in the same way. Regardless, bringing in additional literature, as suggested by the reviewer and clarifying this point – since the PTR-MS measurements were not intended to be dismissed – was done. The paragraph was revised and the discussion extended to address the points of comparison and the possibility of systematic bias (none was identified). The revised text now reads: "Correlations between the canister samples and PTR-MS results were carried out for 35 individual m/z values for which at least one compound was quantified in the canister samples. While the absolute r values of the correlations ranged from 0.00016 to 0.63, the correlations were generally quite poor, showing little to no correlation for many of the m/z (only 9 of the 35 total number of m/z values evaluated had r values greater than 0.3), with no systematic bias identified. There are a number of reasons for this, beyond the difference in how the instruments measure (m/z vs compounds), such as inlet location and sampling time. Previously, in a targeted inter-comparison experiment where whole air samples (canisters) were compared with online PTR-MS

measurements, differences of as little as 20 s in the sampling intervals contributed to scatter in the comparison of the two measurements that was especially relevant for the more reactive NMVOCs (de Gouw and Warneke, 2006). Additionally, scatter in inter-comparisons between ground-based fast time response and GC-MS systems was found to be typical (Lerner et al., 2017 and references therein). In the context of this study, the measurements should not be considered as an inter-comparison since, as described above, the inlets were approx. 5 meters apart, at different heights above ground level, with one street-side and the other above a measurement container. For these reasons, while both measurements are valid, as this comparison shows, the differences in quantification method, but also importantly instrument location and set-up result in substantial differences in what is being quantified so that the comparison is limited in value."

Specific Comments (4) Line 165. It is not clear here how terpenes are affected by canister transport and storage.

Response: Text was added here to explicitly address terpenes. The text now reads, "Oxygenated compounds differed by up to 10% and terpenes by up to 20% over the same time period (Hengst, 2007)."

(5) Line 435. It is not clear what the "BLUME network" is. Some description about what this measurement is would be helpful.

Response: The BLUME network is described initially in the introduction and used subsequently throughout the paper. This refers to measurements from the existing city monitoring network. As a reminder to the reader we have amended the text to read, "...reported from the BLUME city air quality monitoring network...".

(6) Line 440. It is not clear which instrument is located at street-level. In the following discussion, it is implied that the PTR-MS is at street-level, but not explicitly.

Response: Explicit mention of which instruments are associated to which inlet was added to the text at this location.

(7) Line 464. What is m/z 9? This molecule would be smaller than carbon, so not a VOC.

Response: This is a miscommunication. The 9 here refers to the number of m/z ratios out of a total of 35 that had greater than a certain r value. The text has been amended to clarify this point.

(8) Line 499. There are several points below the 1:1, suggesting increases in mixing ratios. It would be interesting to highlight what these compounds are, and whether their lack of decrease/increase in concentration is consistent with the literature.

Response: As per the reviewer's suggestion we have added a paragraph of text to address these points that suggest increases in mixing ratios. We have also added references to the literature that evaluate trends/changes in NMVOCs, however these are very limited for Europe. The text was added to Section 3.3 as follows: "There are a couple of exceptions in this comparison, where the mixing ratios measured in this campaign stand out as substantially higher than those measured 20 years ago. Considering only those few compounds that have a ratio of 0.6 or less for the average mixing ratio in 1996 relative to that in 2014, the biogenic contributions in Neukölln (isoprene (0.3), methylvinylketone (0.1)) show increases. These increases may be attributable to changes in vegetation around the measurement site. Other NMVOCs, such as cis-2-butene and cyclopentane showed increases for both the urban background site and traffic site (Tiergarten Tunnel vs Frankfurter Allee). Other compounds, such as cis-2-pentene and trans-2-butene (traffic site) and 1,2,3-trimethylbenzene (urban background) showed increases at only the one site type. While the literature on trends of NMVOCs is limited, data from a traffic site in London, a rural background site in the UK, and a remote site in Germany showed that over the period from 1998-2009 all individual NMVOCs evaluated (with the exception of n-heptane at the rural background site) were decreasing, with stronger decreases observed at the traffic site relative to the other site types (von Schneidemesser et al., 2010). Similarly, an evaluation of C2-C8 hydrocarbon data, as total HCs and by compound class, for a number of sites across the UK from 1994-2012, also documented decreases across all compound classes (Derwent et al., 2014). Finally, a broader evaluation of the trends in anthropogenic NMVOC emissions across Europe also documented a decrease between 2003 and 2012 (EEA, 2014, 2016). As such, the existing literature does not provide any detailed documentation that might be able to address the potential increases in those few compounds here where an increase was observed. Furthermore, longer-term sampling may show that the increases documented here do not reflect the long-term trend."

(9) Line 522. Why is limonene not included under the biogenic category when it is measured (Table S1)? Not including limonene might understate the biogenic contribution. It would also help to break down the OH reactivity between isoprene, a-pinene, and b-pinene for the Neukolln and Altlandsberg sites. Some terpenes may be manmade and not biogenic (Derwent et al., 2007).

Derwent, R. G., et al. (2007). "Photochemical ozone creation potentials (POCPs) for different emission sources of organic compounds under European conditions estimated with a Master Chemical Mechanism." Atmospheric Environment 41(12): 2570-2579.

Response: This is in part an error in the text and in part a choice of which compounds to include in the OH reactivity analysis. The error in the text refers to the original line 518 where it was stated that 'The NMVOCs included in each of these categories are provided in Table S1.' which is actually not correct. The earlier reference to the compounds included in the classes where only the SI is referenced is more accurate, since it is actually the explanatory text prior to Table S1, now labeled as Section S1, that lists the more limited set of compounds included, rather than all NMVOCs measured, which is what is represented in Table S1. This has been corrected in the revised manuscript to reference Section S1. The justification for the selection of the compounds was included in section 2.2.1.1, and the compounds were chosen based on the typical subset of NMVOCs that have been included in previous analyses of OH reactivity for comparability. Additional analysis was done to evaluate the contribution of additional biogenics, such as limonene, sabinene, and eucalyptol, however, these compounds were not consistently found in all of the samples, including urban background or urban park samples. An indication of the role of some of these additional species might have was added to the text, at the end of section 3.4, and reads: "Finally, while the 57 NMVOCs included here to calculate OH reactivity were chosen to facilitate comparison to previous studies, a more exhaustive list could change the picture. For example, as mentioned above, the limited number of oxygenated NMVOCs measured would likely lessen the contributions of the other compound classes. As an example, adding six additional oxygenated NMVOCs (propanal, 2-butanol, 1-propanol, butanal, 1-butanol, pentanal) increased the total average OH reactivity between 0.12 s-1 (Plänterwald) to 1.7 s-1 (AVUS Motorway). The percent contribution of these six oxygenated NMVOCs ranges between 2.5% and 9.3% of the new total OH reactivity. In contrast, a similar analysis that included three additional biogenic NMVOCs (limonene, sabinene, eucalyptol) showed much smaller additional reactivity, never more than 0.02 s-1. These compounds also were not consistently present across all samples." Finally, we have added some text to the manuscript to acknowledge the potential anthropogenic source for compounds we have classed as biogenic. The text was added to section 3.4 and reads "While studies have shown that a number of NMVOCs, such as isoprene, or other terpenes can also have anthropogenic sources (Derwent et al., 2007;Reimann et al., 2000), we treat them as biogenic and do not try to tease apart the biogenic vs potential anthropogenic contributions in this context."

(10) Section 3.5.1. While I do not disagree with any of the statements made here, it was not clear by the end of the section what the new insights were. Also, this section could benefit from describing the bulk composition first across all samples, and provide better context for the back-trajectory analysis.

Response: This section has been reorganized to address the points of the reviewer. Please see the track changes version of the manuscript for details on the reorganization. Furthermore, additional text has been added to provide context for the back-trajectory analysis, as follows: "Back trajectories were evaluated to provide information on the origin of the air masses and source-receptor relationships (Stein et al., 2015)." The main insights were the comparison of the values for Berlin to other studies in Europe, indicating that there are significant similarities in e.g. the secondary inorganic contributions to PM as other urban areas in Europe. These results were also included to provide context for the further PM/CMB analysis.

(11) Line 696 – 703. Are the diesel and gasoline vehicle contributions from POA only? If so, a caveat may be warranted here that secondary PM from gasoline and diesel vehicles are not included, which are potentially important sources of PM from transportation (see Gordon et al., 2014).

Gordon, T. D., et al. (2014). "Secondary organic aerosol formation exceeds primary particulate matter emissions for light-duty gasoline vehicles." Atmospheric Chemistry and Physics 14(9): 4661-4678.

Gordon, T. D., et al. (2014). "Secondary organic aerosol production from diesel vehicle exhaust: Impact of after treatment, fuel chemistry and driving cycle." Atmospheric Chemistry and Physics 14(9): 4643-4659.

Response: The source profiles used in the CMB are based on chassis dynamometer tests that evaluated a range of vehicles, driving conditions, etc. and therefore are generally representative of POA. The reviewer makes a good point and this caveat has therefore been integrated into the text at the end of the section referenced here and reads: "Furthermore, it should be noted that the source profiles reflect primary organic aerosol emissions, and therefore the secondary aerosol produced from these vehicular sources, which has been shown to be substantial in many cases, depending on the control technologies in use (Gordon et al., 2014a;Gordon et al., 2014b), is not reflected in these attributions."

(12) Line 709. It is not clear how high concentrations of inorganics support the finding of high amounts of SOA. Please describe in further detail.

Response: This sentence could have been written more explicitly. As the inorganics mentioned are the largest contributors to secondary inorganic PM, their presence/higher concentrations indicate that the air masses have undergone some aging, including secondary aerosol formation. This aging would affect not only inorganic but also organic aerosol components. This has been explicitly indicated now in the sentence. "For all samples, a significant amount of secondary organic aerosol was calculated, 0.87 - 4.4 $\mu$g m-3 (18 - 63%). While this was the contribution to OC, high concentrations of secondary inorganics (sulfate, ammonium, nitrate) support the aging of the air masses and the potential for a significant contribution from secondary aerosol overall."

(13) Line 750. While I do not dispute that biogenic VOCs are reactive and have an outsized contribution on OH reactivity, I believe caveats are needed here that missing VOCs not measured could affect the BVOC contributions presented here.

Response: The reviewers make a good point regarding the limitations of such work. As such we have added text to address this point, as follows: "It should however, be acknowledged that only a subset of the total NMVOCs were measured. If the 'missing' NMVOCs were measured this could influence the results, including the contribution of biogenics and other compound classes to OH reactivity." This added text is in line with and re-iterates the additional detail added in response to the comments about research showing greater flux of oxygenated NMVOCs in urban areas.

(14) Figure 6. It would be helpful to label which sites are traffic-dominated, urban background, and urban park under the name of each site.

Response: The figure has been amended to include these labels. Furthermore, a reference to more detailed site classification information available in Table 2 was added to the figure caption.

Please also note the supplement to this comment:
https://www.atmos-chem-phys-discuss.net/acp-2017-1049/acp-2017-1049-AC1-supplement.pdf

———————————————————

[Figure]

**Supplement:**

[revised manuscript text omitted]

---

## Author Response (AR2)

Dear Dr. Karl,

Thank you for your final comments and handling of the manuscript. Please see below specific responses to each of the points where you have requested changes or revisions.

**Co-Editor Decision: Publish subject to minor revisions (review by editor)** (10 May 2018)

by Thomas Karl

Thank you for submitting a revised manuscript and responding to the reviewer's comments.

Before final publication I suggest a couple of corrections and clarifications.

I would generally recommend to be more specific that the measured OH reactivity is based on the available data from NMVOC mixing ratio measurements which is, by definition, always a lower limit.

*Response: This is a good point to highlight. As such we have added explicit mention of this in*

*Section 2.2.1.1., where the amended sentence now reads: "Furthermore, even if all*

*compounds were included, there would still be missing reactivity that is not captured and*

*because no OH measurements were made, the amount of missing reactivity cannot be reliably*

*quantified, therefore the measured OH reactivity here is a lower limit." Additionally, we*

*added text to section 3.4., where the second sentence of the following text was what was*

*added: "In all cases, including other studies discussed, the values presented are calculated*

*OH reactivity based on measurements of NMVOCs and not OH reactivity that was measured*

*directly. Because the OH reactivity estimates are based on a limited number of NMVOCs, the*

*values presented here are a lower limit."*

Further suggested changes:

929 'the reactivity of measured compounds'

531 change to 'spectrometer'

*Response: Both of these changes have been incorporated as suggested.*

620-622: I suggest to shorten this sentence. The cited voltages are not of great interest from a performance point of view. More important information that should be given here is the E/N

ratio (and / or pdrift and Udrift) at which the experiments were performed.

*Response: The referenced sentence with the cited voltages was removed and replaced with*

*information on the drift tube and detection chamber pressures. "The drift tube pressure*

*(pdrift) was kept between 2.1 and 2.3 mbar with a mean of 2.2 mbar. The detection chamber*

*pressure was kept at $2x10^{-5}$ mbar."*

1230: Correct to: If all missing NMVOCs were measured it could influence our results, including the contribution of biogenics and other compound classes to the calculated OH

reactivity.

*Response: This has been changed as suggested.*

I would also like to point out that a recent publication on urban NMVOC has been highlighted in Science magazine, which could be discussed in context of this study (e.g. introduction)

http://science.sciencemag.org/content/359/6377/760.

*Response: Thank you for this recommendation. We were aware of this paper, but it is a good*

*idea to incorporate it into the introductory text. The second sentence of the following text is*

*the main text that was added, but the surrounding sentences were also modified a bit to better*

[revised manuscript text omitted]